# FOURIER HEAD:
# HELPING LARGE LANGUAGE MODELS
# LEARN COMPLEX PROBABILITY DISTRIBUTIONS

**Nate Gillman**[*,1] , **Daksh Aggarwal**[*,1], **Michael Freeman**[1], **Saurabh Singh**[2], **Chen Sun**[1,2]
[1]Brown University, [2]Google DeepMind

## ABSTRACT

As the quality of large language models has improved, there has been increased interest in using them to model non-linguistic tokens. For example, the Decision Transformer recasts agentic decision making as a sequence modeling problem, using a decoder-only LLM to model the distribution over the discrete action space for an Atari agent. However, when adapting LLMs to non-linguistic domains, it remains unclear if softmax over discrete bins captures the continuous structure of the tokens and the potentially complex distributions needed for high quality token generation. We introduce a neural network layer, constructed using Fourier series, which we can easily substitute for any linear layer if we want the outputs to have a more continuous structure. We perform extensive analysis on synthetic datasets, as well as on large-scale decision making and time series forecasting tasks. We also provide theoretical evidence that this layer can better learn signal from data while ignoring high-frequency noise. All of our results support the effectiveness of our proposed Fourier head in scenarios where the underlying data distribution has a natural continuous structure. For example, the Fourier head improves a Decision Transformer agent's returns across four benchmark Atari games by as much as 377%, and increases a state-of-the-art times series foundation model's forecasting performance by 3.5% across 20 benchmarks unseen during training. We release our implementation at `https://nategillman.com/fourier-head`.

### Fourier Head Learns Higher Quality Densities

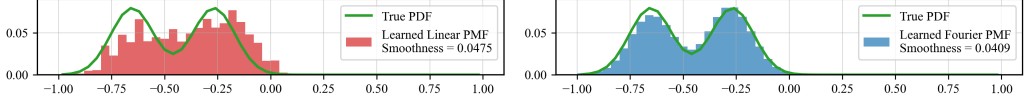

Figure 1: We task an MLP with learning to approximate a continuous bimodal density using a categorical distribution and a cross-entropy objective. We observe that a standard linear head fails to distinguish between the two modes, and overfits to high-frequency noise in the training set. In contrast, our proposed Fourier head learns a smoother, more accurate categorical distribution.

## 1 INTRODUCTION

Human language can be viewed as a discretization for a *continuous*, often *probabilistic* representation of the world that is construed in our mind (Spivey, 2008). The continuous structure can be partially captured by language models with their token embeddings, where "nearby" tokens are embedded to have latent representations with high cosine similarities. The embeddings themselves are acquired as a result of the data-driven learning process. Can we, based on rich prior knowledge

---

*Equal contribution. Correspondence to: nate_gillman@brown.edu, chensun@brown.edu.

about the continuous world, inform the language model about the underlying continuity of its inputs, like the fact that the word "emerald" is more similar to "shamrock" than "pine" when they are used to describe different shades of green? As large language models (LLMs) have evolved into "foundation models" that are adapted to a diverse range of tasks, tokens that are *a priori* continuous are more essential than ever, for example for arithmetic computations (Liu et al., 2023), decision making with continuous or discrete actions (Chen et al., 2021), future anticipation and time-series forecasting (Ansari et al., 2024), or simply drawing random numbers given a probability distribution (Hopkins et al., 2023).

We view the problem of informing LLMs to utilize the continuity prior from the perspective of probability density estimation. For simplicity, we adopt the standard next token prediction framework whose training objective is softmax cross-entropy. Assuming non-overlapping vocabulary, continuous values can be discretized via binning (Ansari et al., 2024). On one hand, the linear head adopted by LLMs independently projects each token into probabilities, and has the expressive power to flexibly approximate arbitrary probability density functions subject to the "quantization" errors. The linear head however does not consider any continuous structure that resides among the tokens (i.e. a random re-shuffle of the tokens in the vocabulary would not change the predictions). On the other hand, a head based on a parameterized distribution (e.g. Gaussian or Gaussian Mixtures) naturally incorporates the continuous structure, but is often too simple (and overly "smooth") to account for multi-modal distributions for future prediction or decision making. Can we design a head that is both expressive and incorporates continuous structures?

We introduce the Fourier head, motivated by Fourier series as universal function approximators. **The Fourier head learns a *continuous* probability density function, and returns a *discrete* approximation of it.** Intuitively, returning a discretization of a continuous density in this way allows the classification head to better model the low-frequency signals from the training data, because the Fourier head is forced to approximate the categorical distributions using a finite number of frequencies. At a high level, the Fourier head inputs $x \in \mathbb{R}^n$, uses a linear layer to learn the coefficients for a Fourier series with $N$ frequencies over $[-1, 1]$, and quantizes the interval $[-1, 1]$ into $m$ equal bins. Then, the Fourier head evaluates the learned Fourier PDF at those $m$ bin center points, and returns those $m$ likelihoods as a categorical distribution. The Fourier head builds upon the Fourier Basis Density Model (De la Fuente et al., 2024).

*Our main contributions are as follows.*

**Contribution #1**: We reveal the underlying principle on the trade-off between the Fourier head's expressive power and the "smoothness" of the predicted distributions. We prove a theorem which demonstrates a scaling law for the Fourier head. Namely, as we increase the quantity of Fourier coefficients learned by the Fourier head, the layer can model increasingly complicated distributions; however, the Fourier head will necessarily fit to more high-frequency noise, thereby outputting categorical distributions which are less smooth.

**Contribution #2**: We propose a practical implementation of the Fourier head capable of sequential prediction tasks by modeling complex multi-modal distributions. Additionally, we propose strategies to improve the layer's performance, including Fourier coefficient norm regularization, weight initialization, and the choice of how many Fourier frequencies to use.

**Contribution #3**: We demonstrate the effectiveness of the Fourier head on two large scale tasks, where intuitively a continuity inductive bias over the output dimensions ought to help the model's generation performance. In the first task, an offline RL agent which uses a decoder-only transformer to model the next-action distribution for an Atari agent, we improve returns across four benchmark games by as much as 377%. In the second, we outperform a state-of-the-art time series foundation model on zero-shot forecasting by 3.5% across a benchmark of 20 datasets unseen during training.

## 2 FOURIER HEAD

### 2.1 FOURIER HEAD: MOTIVATION

When practitioners apply LLMs to model complex probability distributions over non-linguistic tokens, a standard technique is to quantize the latent space into $m$ tokens and learn a conditional categorical distribution over those tokens. We share two examples here:

**Example 1:** The Decision Transformer (Chen et al., 2021) models an Atari agent's behavior in the Seaquest game by learning a categorical distribution over the 18 possible actions (move left, move right, shoot left, etc.). They use an decoder-only transformer architecture.

**Example 2:** The Chronos time series foundation model (Ansari et al., 2024) models the distribution of next numerical values by quantizing the closed interval $[-15, 15]$ into 4096 bins, and learning a categorical distribution over those bins. They use an encoder-decoder transformer.

In a pure language modeling task, token ID 1000 and token ID 1001 likely represent unrelated words. However, in a task where the token IDs represent numerical values, the token ID 1000 and 1001 would represent numbers that are close together.

The final layers of an LLM for such a task are generally a linear layer, followed by softmax, followed by cross-entropy loss. We hypothesize that in scenarios where nearby token IDs encode similar items, an inductive bias that encourages them to have similar probabilities will improve performance. A generic linear layer learns an unstructured categorical distribution and thereby allows more arbitrary probabilities. **In this work, we propose to give the model this inductive bias by letting the classification head learn a categorical distribution as the discretization of a continuous learned function from a suitably flexible class.** We consider the very flexible class of truncated Fourier series with $N$ frequencies. These are functions of the form

$$f(x) = a_0 + \sum_{k=1}^{N} \big(a_k \cos(k\pi x) + b_k \sin(k\pi x)\big). \tag{2.1}$$

Fourier series are a classical tool for solving quantitative problems (Stein & Shakarchi, 2003) because functions like Equation 2.1 are universal function approximators, with the approximation improving as $N$ increases.

## 2.2 Fourier Head: Definition

We now propose a replacement for the generic linear layer token classification head, built using Fourier series. We call our replacement the **Fourier Series Classification Head**, or the **Fourier head** for short. The Fourier head inputs any vector $x \in \mathbb{R}^n$, and outputs a categorical distribution in $\mathbb{R}^m$. For a high level summary of how it works–*the Fourier head inputs $x \in \mathbb{R}^m$, uses a linear layer to extract the coefficients for a Fourier series over $[-1, 1]$, quantizes the interval $[-1, 1]$ into $m$ equal bins, evaluates the learned Fourier PDF at those $m$ bin centerpoints, and returns those $m$ likelihoods as a categorical distribution.* We formally define this layer in Algorithm 1. The Fourier head is constructed using the Fourier Basis Density Model from (De la Fuente et al., 2024). For more details on the original method (e.g. justification for how learning the autocorrelation coefficients guarantees that the Fourier series has integral 1, and justification for normalizing the Fourier coefficients by $\Re(c_0)$) we refer the author to (De la Fuente et al., 2024). We direct the curious reader to Appendix C.1 for a low-dimensional demonstration of the Fourier head in action.

## 2.3 Fourier Head: Considerations for Training

We highlight the main design choices of the Fourier head so that users may apply it most effectively.

**Training objective:** The Fourier head inputs a signal $x \in \mathbb{R}^n$ and extracts from that signal an intermediate representation of a probability distribution $p_x(z)$ defined over $z \in [-1, 1]$. This probability distribution has a closed formula equal to a Fourier series. In our experiments, we optimize the parameters of the Fourier PDF by discretizing it over the latent space and training using cross-entropy loss. However, we should note that the Fourier layer allows MLE training directly on continuous values, by evaluating the Fourier PDF directly on the ground truth value in the latent space. But for consistency of comparison, and to demonstrate how easy it is to swap the Fourier head with a linear layer, we use softmax cross-entropy loss as the objective.

**Choice of hyperparameter $N$:** The Fourier head has one crucial hyperparameter–namely, the number of frequencies. How should one choose this in practice? We offer Theorem 3.3 as guiding principle beyond simple trial and error. This result provides a scaling law which formalizes the smoothness-expressive power trade-off in choosing the number of frequencies. In general, using more frequencies leads to more expressive power, and generally better success metrics, but at the cost of a learning less smooth densities, as well as more model parameters.

---

**Algorithm 1** Fourier head

**Hyperparameters:** the input dimension $n$, output dimension $m$, number of frequencies $N$

**Initialization:** define a linear layer $A : \mathbb{R}^n \to \mathbb{R}^{2(N+1)}$ // maps input to autocorrelation coefficients

$\quad$ INPUT $x = (x_1, \ldots, x_n) \in \mathbb{R}^n$

$\quad (\alpha_0, \beta_0, \ldots, \alpha_N, \beta_N) \leftarrow Ax$

$\quad a_k \leftarrow \alpha_k + i\beta_k \in \mathbb{C}$, for every $k = 0, \ldots, N$ $\qquad$ // compute autocorrelation coefficients

$\quad c_k \leftarrow \sum_{\ell=0}^{N-k} a_\ell a_{\ell+k}^* \in \mathbb{C}$, for every $k = 0, \ldots, N$ $\quad$ // compute Fourier coefficients

$\quad p(z) = \frac{1}{2} + \Re\left(\sum_{k=1}^N \frac{c_k}{\Re(c_0)} \exp(ik\pi z)\right)$ $\qquad$ // define Fourier PDF over $[-1, 1]$

$\quad b_k \leftarrow -1 + \frac{1+2k}{m}$, for every $k = 0, \ldots, m-1$ $\qquad$ // define $m$ bin centerpoints

$\quad y_k \leftarrow \frac{p(b_k)}{\sum_{j=0}^{m-1} p(b_j)}$, for every $k = 0, \ldots, m-1$ $\qquad$ // evaluate PDF at $m$ bin centerpoints

$\quad$ OUTPUT $(y_1, \ldots y_m) \in \mathbb{R}^m$ $\qquad$ // by design, $\sum_{k=1}^m y_k = 1$ and each $y_k \geq 0$

---

**Fourier regularization:** For a given number of frequencies $N$, there could be many learned Fourier models that fit the given data equally well. To encourage a smoother learned model and penalize unnecessary high frequency content, we follow (De la Fuente et al., 2024) and add a regularization term that measures the total squared variation for the Fourier model, to prevent higher order Fourier coefficients from growing too large during training. This helps ensure that the learned Fourier PDF doesn't overfit to noise in the data, and therefore has a bias towards learning smoother densities. In the notation from Algorithm 1, this means adding a regularization term of $\gamma \cdot \frac{2\pi^2}{m} \sum_{k=1}^m k^2 |c_k|^2$ to the loss function, where $\gamma$ is a hyperparameter. When picking regularization strength, we find that in the low-frequency domain (e.g. frequencies in the single digits) using $\gamma = 0$ works best, and in the high-frequency domain (e.g. greater than 10 frequencies), using $\gamma = 10^{-6}$ works best.

**Binning strategy:** The choice of data binning can impact performance. As discussed, the Fourier head should only be applied when nearby bins are 'similar' in some sense, requiring a semantically meaningful bin ordering. When bins represent quantized numerical values over a continuous latent space, a 'mixed-precision' binning strategy can improve performance. For example, to model values in $[-15, 15]$ with most data in $[-1, 10]$, allocating more bins to the dense interval improves performance. Given $m$ total bins, a hyperparameter $d \in [0, 1)$ controls allocation, with $\lfloor d \cdot m \rfloor$ bins for the sparse interval and the rest for the dense range (estimated from training data). Fourier theory supports this approach, as increasing precision in dense regions de-localizes the quantized data distribution, localizing the Fourier spectrum. This accelerates higher frequency decay, enabling effective learning with lower-frequency Fourier heads. Separately, we note that (De la Fuente et al., 2024) suggests re-parameterizing the periodic domain to the real line, though we do not use this in our work.

**Weight initialization:** The learned parameters for the Fourier head consist of the learned linear layer which extracts autocorrelation parameters $a_k$. In PyTorch, linear layers use He initialization (He et al., 2015) by default, which ensures that the linear layer outputs values close to zero in expectation. Similarly, initializing the Fourier densities to be uniform $p(z) \approx 1/2$ improves learning dynamics. We accomplish this by dividing the weights and biases by a large number, such as 1000, after He initialization; this guarantees that the linear layer outputs very small values, so that Fourier coefficients output from the autocorrelation step are very small as well.

## 3 THEORETICAL ANALYSIS OF FOURIER HEAD

### 3.1 "SMOOTHNESS": A METRIC FOR HIGH FREQUENCY CONTENT

In this subsection we propose a smoothness metric which inputs a categorical distribution $y = (y_1, \ldots, y_m) \in \mathbb{R}^m$, and assigns a numerical value depending on how smooth it is. The score will output 0 if $y$ is the smoothest possible categorical distribution, and larger values if $y$ is less smooth. We will first specify what we mean by "smooth":

**Heuristic 3.1.** *We say a function is **smooth** if it contains **very little high-frequency information**.*

For example, the uniform categorical distribution contains no high-frequency information, so it is the smoothest possible function, and should get a smoothness score of $0$. In contrast, a categorical distribution containing samples from $\sin(100\pi x)$ contains lots of high frequency information, so it should get a smoothness score greater than $0$. We seek to define a metric which measures smoothness according to Heuristic 3.1.

We will first develop a smoothness metric in the general case of a function $f : [a, b] \to \mathbb{R}$, then specialize to case of the discrete categorical distribution that we consider in the paper. If we let $\alpha_\sigma \in \mathbb{R}$ be weights satisfying $\int_0^\infty \alpha_\sigma d\sigma = 1$, and $D$ be some measure of discrepancy such as $L^2$, and let $g_\sigma(x) * f(x)$ denote the convolution of $f(x)$ with a Gaussian kernel of standard deviation $\sigma$, then it is reasonable to define the smoothness of $f$ to be the quantity

$$s(f) := \int_0^\infty \int_a^b \alpha_\sigma D[f(x), g_\sigma(x) * f(x)] dx d\sigma. \tag{3.1}$$

In this expression, the discrepancy $D[f(x), g_\sigma(x) * f(x)]$ measures how different $f(x)$ is from a Gaussian-smoothed version of itself. Because the Gaussian is a low-pass filter, we can interpret Equation 3.1 as saying, at a high level, that a function is "smooth" if it doesn't change that much when you remove high frequency content from it.

In our experiments, we consider discrete categorical distributions, and wish to tractably quantify their smoothness. Accordingly, we define a specific case of this as follows.

**Definition 3.2** (Smoothness metric for categorical distributions). *Suppose $y = (y_1, \ldots, y_m) \in \mathbb{R}^m$ is a categorical distribution, so every $y_k \geq 0$ and $\sum_{k=1}^m y_k = 1$. Denote by $g_\sigma \in \mathbb{R}^{2m-1}$ the discrete Gaussian kernel of standard deviation $\sigma$ and radius $m - 1$. Define the weights $\alpha_\sigma = 6/\pi^2\sigma^2$. Then we define the **smoothness** of $y$ to be the constant*

$$s(y) := \sum_{\sigma=1}^\infty \alpha_\sigma \|y - g_\sigma * y\|_2. \tag{3.2}$$

We direct the curious reader to Appendix B, where we conduct additional experiments to justify this choice of smoothness metric for our experiments.

### 3.2 A Scaling Law for the Fourier Head, in Frequency-aspect

In this subsection, we share a theorem that analyzes the quality of the Fourier head as the quantity of frequencies changes. We refer to this as the **Fourier head scaling law** as it quantifies the trade-off between modeling capacity and smoothness as the number of frequencies increases. On one hand, it is a celebrated result from Fourier analysis that a Fourier series with a greater number of frequencies models a larger class of functions; but on the other hand, we show that increasing frequencies also incurs loss in smoothness. This is to be expected, as we designed our smoothness metric with the intention of identifying a distribution as less smooth if it contains more high-frequency information.

**Theorem 3.3.** *(Fourier head scaling law.) Consider a Fourier head with input dimension $n$, output dimension $m$, and $N$ frequencies. Suppose that $1 \ll N < \frac{m}{2}$. Then the following are true:*

1. *(**Increasing $N$ improves modeling power**.) As $N$ increases, the Fourier head is capable of learning a larger class of densities.*

2. *(**Increasing $N$ degrades smoothness**.) Consider an input to the Fourier head $x \in \mathbb{R}^n$, and denote by $f_x : [-1, 1] \to \mathbb{R}$ the optimal conditional distribution that we would like the Fourier head to approximate for this input. Suppose that there exists some $t \geq 2$ such that the Fourier coefficients of $f_x$ decay on the order of $1/k^t$. Denote by $f_{x,N}$ the truncation of $f_x$ to its first $N$ frequencies, denote by $\vec{b} \in \mathbb{R}^m$ the $m$ bin centerpoints in $[-1, 1]$, and denote by $y^{(N)} = f_{x,N}(\vec{b})/(f_{x,N}(b_0) + \cdots + f_{x,N}(b_{m-1})) \in \mathbb{R}^m$ the discretization of $f_{x,N}$ into $m$ bins. Then, there exist constants $C_1, C_2 > 0$ such that*

$$s(y^{(N)}) = C_1 - \frac{C_2}{N^{2t-1}} + O(1/N^{2t}). \tag{3.3}$$

Toy Example: Learned Conditional Distribution vs True Conditional Distribution

Figure 2: Comparison between the PMFs learned by the linear head, GMM head, and the Fourier head, for two of the datasets in the toy example–Gaussian and Beta. (The GMM dataset is in Figure 1.) We observe that the Fourier head learns a smoother categorical distribution than the linear head over its predicted values. Furthermore, the Fourier head better fits the true conditional PDF; this is reflected in the KL divergence and smoothness metrics.

Note that the smoothness scaling law asymptotic in Equation 3.3 shows that as $N$ increases, so does $s(y^{(N)})$. Further, note that if the Fourier spectrum of the underlying distribution decays quicker (controlled by $t$) then the rate at which smoothness degrades is slower; this is because if what we are learning has little high frequency content, then increasing the frequencies shouldn't affect the smoothness of the learned distribution very much. In part (2), our assumption that the Fourier coefficients decay at least quadratically is reasonable since if $f_x$ is at least twice continuously differentiable, we already know its Fourier coefficients corresponding to the $k$-th frequency are in $O(1/k^2)$ (Stein & Shakarchi, 2003, Ch.2, Cor. 2.4). Our Fourier weight decay regularization helps toward ensuring that this condition is met in practice as well. We include a full proof in Appendix A.

## 4 TOY EXAMPLES

### 4.1 LEARNING A CONTINUOUS CONDITIONAL DISTRIBUTION

We demonstrate the advantage of using the Fourier head to learn a probability distribution for a simple task: learning the conditional distribution of the third number in the sequence given the first two. Here we will use $q(z)$ to denote the quantization of $z$.

**Dataset**: We create 3 synthetic datasets, which we name **Gaussian**, **GMM-2**, and **Beta**. Each dataset consists of 5000 quantized triples $\{(q(x), q(y), q(z))\} \subseteq [-1, 1]^3$. Crucially, $z$ is sampled from a distribution which is conditioned on $x$ and $y$, and we have an explicit closed formula for this distribution. By design, the Gaussian dataset is unimodal in $z$, whereas the more challenging GMM-2 and Beta datasets are not unimodal. Full details about the datasets can be found in Appendix C.2.

**Task**: Predict the conditional distribution of $q(z)$ given the quantized tuple $(q(x), q(y))$.

**Model architecture**: Our model is an MLP with ReLU activations and one hidden layer, which maps $\mathbb{R}^2 \to \mathbb{R}^{64} \to \mathbb{R}^{32} \to \mathbb{R}^{50}$. The output of the model has dimension 50 because we quantize the interval $[-1, 1]$ into 50 bins. We consider two baselines alongside the Fourier model. For the first baseline, the classification head is a linear layer; for the second baseline, the classification head is a Gaussian model mixture classification layer with two Gaussians, where the means and standard deviations are learned; for the Fourier model, the classification head is the Fourier head. We sweep over frequencies $N = 2, 4, \ldots, 20$, and consider regularization $\gamma \in \{0, 10^{-6}\}$. We train those models via cross-entropy loss.[1] We also consider a regression-based model, trained using MSE.

**Model evaluation**: We use three metrics for evaluation. Our first metric is the average KL divergence $D_{\text{KL}}(q(\mathcal{P}(x, y))||M(q(x), q(y)))$, where $\mathcal{P}(x, y)$ is the fixed conditional distribution of $z$ given $(x, y)$; $q(\mathcal{P}(x, y))$ is the quantized approximation of $\mathcal{P}(x, y)$, obtained by evaluating the den-

---

[1]Note that we also demonstrate the possibility of training a *continuous* version of the Fourier head, using a maximum-likelihood based objective. Accordingly, we carry out experiments in the continuous domain analogous to those we did in the quantized domain; for more details, see Appendix C.3.

| Dataset | KL Divergence (↓) | | Smoothness (↓) | |
|---|---|---|---|---|
| | Linear | Fourier | Linear | Fourier |
| Gaussian | $0.170 \pm 0.052$ | $\mathbf{0.116} \pm 0.043$ | $0.116 \pm 0.049$ | $\mathbf{0.057} \pm 0.011$ |
| GMM-2 | $0.238 \pm 0.032$ | $\mathbf{0.146} \pm 0.033$ | $0.068 \pm 0.022$ | $\mathbf{0.038} \pm 0.007$ |
| Beta | $0.234 \pm 0.032$ | $\mathbf{0.191} \pm 0.016$ | $0.127 \pm 0.044$ | $\mathbf{0.076} \pm 0.021$ |

Table 1: We compare metrics between the linear head, and the Fourier head with 12 frequencies and no regularization, for every dataset in our toy example. We observe that the Fourier head outperforms the linear head across all metrics. Notably, using Fourier head improves the KL divergence (the primary success metric) on average by approximately 40%. We aggregate metrics over 4 different seeds and report the standard deviation.

Using Llama-3.1-8B-Instruct to Simulate Gaussian Sampling

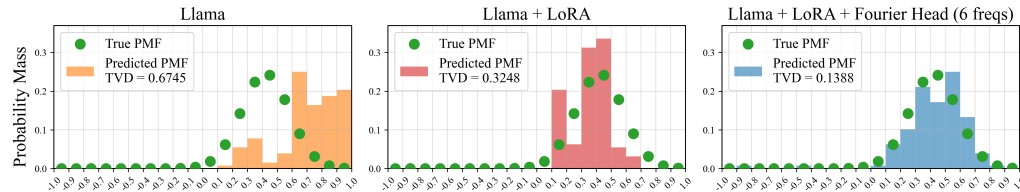

Figure 3: We demonstrate that the baseline Llama model does a poor job simulating Gaussian sampling, as measured by the Total Variation Distance between the ground truth quantized Gaussian histogram, and the empirical histogram of samples. We find that LoRA fine-tuning improves the results by a factor of $\approx 2.07$, and that using the Fourier head improves the output distribution by a factor of $\approx 4.86$.

sity function of $\mathcal{P}(x, y)$ at the bin centers, multiplying by the bin width, and finally scaling by the sum of the likelihoods; and $M(q(x), q(y))$ denotes the predicted categorical conditional distribution of $q(z)$. Our second metric is smoothness. And our third metric is MSE, where we consider the expected value of $q(z)$ under the learned categorical distribution as a prediction for $q(z)$.

**Results:** The metrics for the best performing model on each dataset are reported in Table 1. Figure 2 presents sample visualizations of the learned conditional distributions alongside the true densities. And in Appendix C.2, we present the results of a study on the impact of number of frequencies and Fourier regularization. Notably, this study provides empirical evidence for the Fourier head scaling law in Theorem 3.3, as it demonstrates that for all datasets, as frequency increases, the smoothness degrades, and model performance improves until it reaches a saturation point. Crucially, we observe that the Fourier head flexibly learns all three distributions better than the linear baseline does. We note that the Fourier head outperforms the linear head on MSE as well; we include a complete comparison with both Linear and GMM head baselines in Appendix C.2. Additionally, in Figure 10 (Appendix), we demonstrate that the regression model simply regresses to the mean of the conditional distribution. Accordingly, the regression model performs well for the unimodal Gaussian dataset, and it performs poorly for the bimodal datasets GMM-2 and Beta.

## 4.2 ARE LLMs RANDOM NUMBER GENERATORS?

Suppose that we query an LLM with the following prompt, repeatedly: *"The following is a list of normally distributed random numbers in the interval [-1, 1] with mean 0.55 and std 0.10: 0.57, 0.36, "*. Would the model outputs be approximately Gaussian? In this empirical study, we simulate Gaussian sampling using the Llama-3.1-8B-Instruct model (Dubey et al., 2024). We demonstrate that this language model struggles to generate high quality numerical Gaussian samples. We consider two possible interventions: LoRA fine-tuning the base Llama model, and LoRA fine-tuning the base model while also replacing the linear classification head with a Fourier head. We find that LoRA fine-tuning improves the learned distribution significantly, and replacing the linear head with the Fourier head improves the distributions even further. We present an illustrative example of this phenomenon in Figure 3. See Appendix C.4 for experiment details and some related works.

## 5 LARGE-SCALE STUDY: OFFLINE REINFORCEMENT LEARNING

The Decision Transformer (Chen et al., 2021) reframes reinforcement learning as sequentially modeling rewards, states, and actions. We evaluate its performance on the Seaquest game from the Atari (Bellemare et al., 2013) benchmark. The Seaquest game contains 18 actions, with two groups of eight actions that have a natural "closeness" metric defined on them: move left, up left, up, up right, right, down right, down, down left; as well as shooting in those eight directions. The original architecture uses a decoder-only language model (Radford et al., 2018) to encode context and map it through a linear layer, producing a categorical distribution over actions. At test time, the agent samples from this distribution to select its next action. We replace the linear classification head with a Fourier head, introducing a prior that semantically similar actions (e.g., 'move left' and 'move up left') should have similar likelihoods. Our results show the Fourier head improves returns by as much as $46\%$ in the reward-conditioned setting, using identical training hyperparameters.

**Task**: In the Seaquest game, the agent moves a submarine to avoid enemies, shoot at enemies, and rescue divers. We consider this task in the Offline RL setting. The agent observes the past states, actions, and rewards, as well as the return-to-go, and attempts to predict the action that matches what an agent operating like the dataset would likely do. We also consider three other Atari games with the same action space: BankHeist, DoubleDunk, and Gravitar.

**Dataset**: We use the same dataset from the original Decision Transformer implementation (Chen et al., 2021). This dataset consists of 500k transitions experienced by an online deep Q-network agent (Mnih et al., 2015) during training on the Seaquest game.

**Model architecture**: (Chen et al., 2021) used the GPT-1 model (Radford et al., 2018) to autoregressively encode the context, which is then fed through a linear layer of dimension 18, and the model ultimately optimizes the cross-entropy loss between the action logits and the ground truth action from the dataset. We refer to this model as the linear baseline. To create our Fourier-$N$ version, we simply replace the linear head with a Fourier head with $N$ frequencies and Fourier regularization $\gamma = 10^{-6}$. In our experiments we consider frequencies $N \in \{2, 4, 6, 8, \ldots, 30, 32\}$.

Normalized Returns for Decision Transformer Agent

| | Atari Game | | | |
|---|---|---|---|---|
| Classification Head | BankHeist | DoubleDunk | Gravitar | Seaquest |
| Linear head | $-0.09 \pm 0.05$ | $-72.72 \pm 33.08$ | $1.32 \pm 0.17$ | $2.53 \pm 0.63$ |
| Fourier head | $\mathbf{0.92 \pm 0.33}$ | $\mathbf{45.45 \pm 36.36}$ | $\mathbf{4.98 \pm 0.93}$ | $\mathbf{3.70 \pm 0.47}$ |

Table 2: We present returns obtained by the Decision Transformer agent using the linear baseline, and the Fourier head, across the four Atari games. We compute the returns (mean and standard deviation) by averaging over four seeds. Across all these games, the Fourier head significantly improves the normalized returns obtained by the agent.

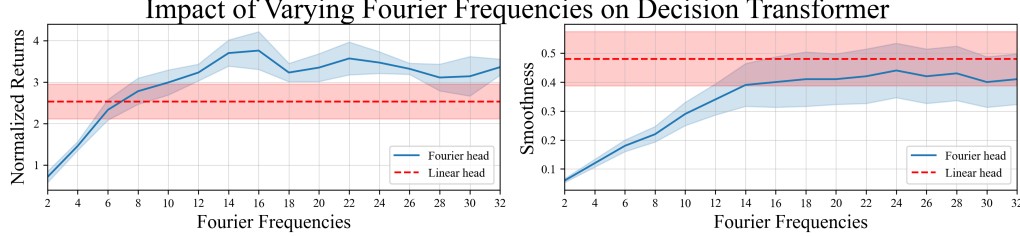

Figure 4: We present empirical results for how the quantity of Fourier frequencies impacts returns and smoothness for the imitation learning task. For normalized returns, higher is better; for smoothness, lower is better. We can see that the Fourier agent achieves higher normalized returns than the linear baseline agent when sufficiently many Fourier frequencies are used, while still learning smoother next-action distributions.

**Model Evaluation**: We present results for the linear baseline and Fourier-$N$ head ($N \in 2, 4, 6, 8, \ldots, 30, 32$) across four Atari games, showing mean reward totals for rollouts at the best epoch across four seeds. Table 2 demonstrates significant return gains with the Fourier head. For example, Seaquest returns increase by up to $46.2\%$, while Gravitar sees as much as a $377\%$ boost. Figure 4 shows improved Seaquest performance as the number of frequencies grows, with learned PMFs becoming less smooth, aligning with Theorem 3.3. Qualitative results in Figure 13 (Appendix) highlight the smoother PMFs produced by the Fourier head. Additional results for BankHeist, Double-Dunk, and Gravitar in Figure 16 (Appendix) confirm that the Fourier agent consistently outperforms the linear baseline while maintaining smoother next-action distributions.

**Ablations**: We analyze whether model size has any effect on the relative performance of the Linear head and the Fourier head. The results in Figure 14 (Appendix) demonstrate that, across model sizes, the Decision Transformer with a Fourier head is better at learning high-quality next action distributions than the Decision Transformer with a Linear head. We also analyze whether dataset size has any effect on the relative performance of the Linear head and the Fourier head, and obtain a similar result. In Figure 15 (Appendix) we show that, across dataset sizes, the Decision Transformer agent with the Fourier head achieves larger returns than the agent with a linear head.

## 6 LARGE-SCALE STUDY: PROBABILISTIC TIME SERIES FORECASTING

The Chronos time series foundation models (Ansari et al., 2024) "learn the language of time series". They do this by approaching time series forecasting as language modeling, by tokenizing the quantized number line, learning token embeddings for each of those quantized values, and finally learning a categorical distribution to decide what the next value ought to be. This model is built on top of the encoder-decoder T5 model (Raffel et al., 2020). In particular, this model normalizes time series values to the range $[-15, 15]$ and quantizes this interval into 4096 tokens. As usual for language modeling, the final layer is a linear map which learns a categorical distribution over next tokens. In particular, we observe that token $i$ represents a number very close to tokens $i - 1$ and $i + 1$. However, we note that there is no inductive bias in the T5 architecture which pushes their likelihoods to be similar. This is not a hypothetical problem; in Figure 17 (Appendix), we can see that the linear next-token prediction PMFs fit to the noise, and appear very jagged.

**The motivation for replacing the linear head with the Fourier head is to "smooth" out the distribution in the left side of Figure 17, to help the forecasting model better learn the signal, and ignore the noise.** In this figure, we can see that the Fourier head accomplishes this successfully.

In this section, we study how the performance of the Chronos time series foundation model changes when we pre-train using the Fourier head, instead of the linear head. For all of the frequencies that we consider, the Fourier head outperforms the Chronos linear baseline on the MASE metric, while learning next token multinomials which are at least 8x smoother, with fewer parameters than the baseline.

**Dataset**: We use the same training dataset for large-scale pretraining that Ansari et al. (2024) used. We gather an evaluation benchmark of 20 time series datasets which were not seen during training. These 20 come from the zero-shot eval from (Ansari et al., 2024). The reader can check Appendix D.2 for details on the training and evaluation datasets we used.

**Model architecture**: We use the Chronos model, which is built using the T5 architecture (Raffel et al., 2020). The original model has a linear classification head. For our study, we will replace this with a Fourier head with frequencies $N = 64, 128, 256, 550$. We use mixed precision binning; this is informed by an analysis of the Fourier spectrum of the next-token distribution, as described in Section 2.3. We also use Fourier weight decay regularization with $\gamma = 10^{-6}$. For the task, the model

| Chronos Time Series Model | MASE ($\downarrow$) | WQL ($\downarrow$) | Smoothness ($\downarrow$) |
|---|---|---|---|
| Linear | 0.883 | 0.750 | $0.1689 \pm 0.1087$ |
| **Fourier-550** | **0.852** | **0.749** | **$0.0283 \pm 0.0224$** |

Table 3: We present large-scale experiments on Chronos time series forecasting. The best-performing Fourier model outperforms the linear baseline both terms of the continuity of the learned probability mass functions (smoothness) for the quality of the forecasts (MASE, WQL).

learns to input time series context of length 512, and output a probabilistic forecast of length 64. At test time, the model chooses the next numerical token by sampling from the next-token distribution

**Model evaluation**: We have two sets of metrics: model performance from (Ansari et al., 2024) (MASE measures the accuracy of median forecast, and WQL measures the quality of the probabilistic forecast), as well as our smoothness metric. Our Fourier metrics in Table 3 demonstrate that every Fourier model outperforms the linear baseline for MASE and smoothness. Furthermore, for the largest Fourier model that we consider, Fourier outperforms linear on WQL as well.

**Ablations:** The results in Table 8 (Appendix) show that mixed precision binning and regularization improve the MASE and smoothness for the Fourier head, and that using more Fourier frequencies improves MASE and WQL. Additionally, we show that the Fourier head yields more accurate forecasts than the linear head across dataset sizes and model sizes (Figures 18 and 19, Appendix).

## 7    RELATED WORK

**LLMs outside of natural language domains:** LLMs are often adapted to domains beyond natural language, as general purpose sequence models. For example, they have been used in protein synthesis (Madani et al., 2023), time series forecasting (Ansari et al., 2024; Das et al., 2024; Jin et al., 2024; Nate Gruver & Wilson, 2023; Requeima et al., 2024; Jia et al., 2024; Zhou et al., 2023; Wang et al., 2024), music generation (Dhariwal et al., 2020; Agostinelli et al., 2023; Copet et al., 2023; Yuan et al., 2024), and as well as in decision making (Li et al., 2022; Chen et al., 2021).

We consider three categories to adapt LLMs to non-language domains: when the output of a language-trained LLM is used as a feature for some out-of-domain task; when a language-pretrained LLM is fine-tuned on a domain-specific task; and when an LLM architecture is trained on a domain-specific dataset from scratch. Our work directly considers the latter method of LLM adaptation, particularly in settings where the outputs approximate continuous values. We note that using LLMs to model numerical functions has seen success in continuing sequences (Mirchandani et al., 2023) but has been challenging for modeling samplers for probability distributions (Hopkins et al., 2023). In a related direction, Razeghi et al. (2022) found that model performance on numerical reasoning tasks is correlated with the frequency of specific numbers in its corpus. Further, some have re-framed continuous regression as a descretized classification problem to leverage LLMs in numerical modeling contexts (Song et al., 2024) or RL contexts (Farebrother et al., 2024). While even frozen LLMs with no further training show interesting empirical results as regressors (Vacareanu et al., 2024), there is a conceptual mismatch between the downstream task and model construction because tokenized numerical values trained using cross-entropy loss does not explicitly enforce numerical relationships between the tokens.

**Fourier series in neural networks:** Many works leverage the Fourier transform as a data pre-processing step or a deterministic transformation within the network, or use Fourier analysis to motivate design choices. It is far less common to learn the Fourier series directly. De la Fuente et al. (2024) learned marginal univariate densities parameterized using a Fourier basis; our work extends their Fourier Basis Density model to multivariate settings with an autoregressive scheme. Our method learns conditional univariate densities using a Fourier basis, where the coefficients of the Fourier density model are input dependent. Sitzmann et al. (2020) proposed sinusoidal activation functions, which can be seen as learning the *frequencies* of a Fourier series; in contrast, we fix the frequencies to the canonoical choice $\{1, 2, \ldots, N\}$, and learn the *amplitudes*. This allows the Fourier head to more directly benefit from approximation results from Fourier analysis.

## 8    CONCLUSION

We propose the Fourier head and demonstrate its positive impact on performance on several tasks. We prove a scaling law that characterizes the trade-off between the model's expressivity and the smoothness of its output distribution. The Fourier head is a modular architecture that can be easily added to existing models that would benefit from the continuity inductive bias that the head imparts. The Fourier head extends the already extensive reach of LLMs into more diverse, numerical, and probabilistic domains. Future work includes exploring alternative training objectives that do not depend on discretizing probability density functions, and incorporating the Fourier head in general-purpose LLM training, where the head can be adaptively employed when needed.

## 9 REPRODUCIBILITY STATEMENT

We have made efforts to ensure reproducibility. In Algorithm 1 we provide all the mathematical details that one needs to reproduce the Fourier head. In Appendix D.2 we prove our scaling law, Theorem 3.3, in full detail, and we list all assumptions in the statement of the theorem. Additionally, we have released the research code on GitHub: `https://github.com/nate-gillman/fourier-head`.

### ACKNOWLEDGMENTS

We would like to thank Jona Balle, Alfredo De la Fuente, Calvin Luo, Singh Saluja, Matthew Schoenbauer, and Megan Wei for the useful discussions. This work is supported by the Samsung Advanced Institute of Technology, NASA, and a Richard B. Salomon Award for Chen Sun. Our research was conducted using computational resources at the Center for Computation and Visualization at Brown University. Chen would like to thank Mia for inspiration.

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

# Appendix

## Table of Contents

## A  PROOF OF FOURIER HEAD SCALING LAW, THEOREM 3.3

In this section we prove Theorem 3.3, the Fourier head scaling law. To do this, we must first discuss the Nyquist-Shannon Sampling Theorem. This result states that in order to avoid distortion of a signal (such as aliasing) the sampling rate must be at least twice the bandwidth of the signal. In the setting of the Fourier head, our sampling rate is $m/2$ because we have $m$ bins uniformly spaced in $(-1, 1)$, and the bandwidth is $N/2$ because the frequency of $\sin(\pi N x)$ is $N/2$. Thus the Nyquist Theorem requires us to have

$$m/2 \geq 2 \cdot (N/2) = N$$

in order for the higher order frequency content learned by our model to not be fallacious when we are learning from only $m$ bins. This justifies why we only theoretically study the case $1 \ll N < m/2$ in the scaling law.

### A.1  DEFINITIONS

Consider an input $x \in \mathbb{R}^n$ to the Fourier head, and denote by $f_x : [-1, 1] \to \mathbb{R}$ the optimal conditional distribution that we would like the Fourier head to approximate for this input. We will assume that $f_x$ is periodic, since the Fourier head learns a 2-periodic Fourier density. We denote by $f_{x,N}$ the truncation of the Fourier series of $f_x$ to its first $N$ frequencies. Note that $f_{x,N}$ also integrates to 1 over $[-1, 1]$ since its first Fourier coefficient is the same as that of $f_x$. Further, $f_{x,N}$ is non-negative on $[-1, 1]$ since its Fourier coefficients, being a subsequence of the coefficients of $f_x$, are non-negative definite; a periodic function with non-negative definite Fourier coefficients is non-negative by Herglotz's Theorem (Brockwell & Davis, 1991, Corollary 4.3.2). For completeness, we will recall the convolution formulas, specialized to the cases we consider in our argument.

**Definition A.1** (Discrete convolution). *Let $b_j := -1 + \frac{2j+1}{m}, 0 \leq j < m$ be the center points of the $m$ bins in $(-1, 1)$, and let us denote $\vec{b} := (b_0, \ldots, b_{m-1})$. Denote by $G_\sigma(z) := \frac{e^{-z^2/2\sigma^2}}{\sqrt{2\pi}\sigma}$ the Gaussian PDF with standard deviation $\sigma$. Then the discrete Gaussian convolution filter of radius $m - 1$ is*

$$g_\sigma := \frac{G_\sigma([1-m, 2-m, 3-m, \ldots, m-1])}{S(m, \sigma)} \in \mathbb{R}^{2m-1}, \tag{A.1}$$

*where the normalization constant is*

$$S(m, \sigma) := \sum_{k=1-m}^{m-1} G_\sigma(k). \tag{A.2}$$

*The discrete convolution of $g_\sigma \in \mathbb{R}^{2m-1}$ and $f_{x,N}(\vec{b}) \in \mathbb{R}^m$ is the vector $(g_\sigma * f_{x,N})(\vec{b}) \in \mathbb{R}^m$ whose $j$'th coordinate is given by*

$$(g_\sigma * f_{x,N})(b_j) = \frac{1}{S(m, \sigma)} \sum_{k=1-m}^{m-1} G_\sigma(k) \cdot f_{x,N}(b_{j-k}). \tag{A.3}$$

**Definition A.2** (Continuous convolution). *The continuous Gaussian convolution filter $\tilde{g}_\sigma : [-2, 2] \to \mathbb{R}_{>0}$ is*

$$\tilde{g}_\sigma(z) = \frac{G_{\frac{2\sigma}{m}}(z)}{S(m, \sigma)} = \frac{m}{2S(m, \sigma)} G_\sigma\left(\frac{mz}{2}\right). \tag{A.4}$$

*This function $\tilde{g}_\sigma(z)$ is a normalized truncation of a Gaussian PDF with mean $0$ and standard deviation $2\sigma/m$. The continuous convolution of $\tilde{g}_\sigma : [-2, 2]$ and the periodic function $f_{x,N} : [-1, 1] \to \mathbb{R}$ is*

$$\tilde{g}_\sigma * f_{x,N}(z) := \int_{-2}^{2} \tilde{g}_\sigma(u) f_{x,N}(z - u) \, du. \tag{A.5}$$

## A.2 OVERVIEW OF PROOF

In this subsection, we provide an overview of the proof of Theorem 3.3 by presenting the statements of the lemmata that we will need, and connecting each one to the overall argument. In the next subsection, we rigorously prove the scaling law by careful applications of these lemmata. And in the following subsection, we will rigorously prove each of the lemmata.

This first lemma allows us to replace the discrete Gaussian convolution in the definition with a continuous Gaussian convolution.

**Lemma A.3.** (Discrete convolution is close to continuous convolution) *If we define the constant $B_1(m, \sigma) := 1 + \frac{G_\sigma(m)}{S(m,\sigma)}$, then we have that*

$$\|f_{x,N}(\vec{b}) - g_\sigma * f_{x,N}(\vec{b})\|_2 = \|B_1(m, \sigma) f_{x,N}(\vec{b}) - \tilde{g}_\sigma * f_{x,N}(\vec{b})\|_2 + \sqrt{m} O(1/N^{2t+1}). \tag{A.6}$$

*Furthermore, $B_1(m, \sigma)$ satisfies the following bound, uniformly in $\sigma$,*

$$B_1(m, \sigma) \le 1 + \frac{1}{2m - 1}. \tag{A.7}$$

This next lemma, a standard result from analytic number theory allows us to upper bound the sums of the norms of the Fourier series coefficients. This is proved in various places, see e.g. (Tao, 2014, Equation 21).

**Lemma A.4** (Asymptotic expansion of Riemann zeta function). *Consider the Riemann zeta function $\zeta(t) := \sum_{k=1}^{\infty} \frac{1}{k^t}$. If $t \ge 2$, then*

$$\sum_{k=1}^{N} \frac{1}{k^t} = \zeta(t) - \frac{1}{t - 1} \frac{1}{N^{t-1}} + O(1/N^t). \tag{A.8}$$

This next lemma allows us to extract the main asymptotic behavior in the scaling law.

**Lemma A.5.** (Main term asymptotic) *Denote by $a_0(x)$ the constant coefficient of $f_{x,N}$. Let us suppose that the Fourier coefficients of $f_{x,N}$ decay like $B_3(x)/k^t$, and define the constant*

$$B_2(\sigma, m, x) := \sqrt{a_0(x)^2 B_1(m, \sigma)^2 + 2 B_1(m, \sigma)^2 B_3(x)^2 \zeta(2t)}. \tag{A.9}$$

*Then we know that*

$$\|B_1(m, \sigma) f_{x,N}(\vec{b}) - \tilde{g}_\sigma * f_{x,N}(\vec{b})\|_2 \tag{A.10}$$

$$= \sqrt{m} \left( B_2(\sigma, m, x) - \frac{B_1(m, \sigma)^2 B_3(x)^2}{2t - 1} \cdot \frac{1}{N^{2t-1}} + O(1/N^{2t}) \right). \tag{A.11}$$

*Furthermore, $B_2(\sigma, m, x)$ is bounded from above and below as a function of $\sigma$ and $m$.*

This final lemma allows us to relate the continuous case, where our analysis works out easier, to the discrete case, where our smoothness metric is actually defined.

**Lemma A.6.** (The average value of the truncated Fourier PDF is $1/2$) *If $N < m/2$, then*

$$\sum_{j=0}^{m-1} f_{x,N}(b_j) = \frac{m}{2}. \tag{A.12}$$

### A.3 PROVING THEOREM 3.3 USING THE LEMMATA

We now prove the theorem that provides a scaling law for the Fourier head. This result quantifies the trade-off between modeling capacity and smoothness as the number of frequencies increases. In order to prove this, we must assume that $f_x$, the conditional distribution being learned by the Fourier head, is sufficiently smooth. For example, if $f_x$ is twice continuously differentiable, then the Fourier coefficients corresponding to the $k$-th frequency of $f_x$ are in $O(1/k^2)$ (Stein & Shakarchi, 2003, Ch.2, Cor. 2.4). Thus, our assumption that the Fourier coefficients decay quadratically is reasonable, and our Fourier weight decay regularization helps ensure that this condition is met in practice as well. In our theorem, we generalize this hypothesis to the cases where the Fourier coefficients corresponding to the $k$-th frequency of $f_x$ are in $O(1/k^t)$.

**Theorem 3.3.** *(Fourier head scaling law.) Consider a Fourier head with input dimension $n$, output dimension $m$, and $N$ frequencies. Suppose that $1 \ll N < \frac{m}{2}$. Then the following are true:*

1. *(**Increasing $N$ improves modeling power**.) As $N$ increases, the Fourier head is capable of learning a larger class of densities.*

2. *(**Increasing $N$ degrades smoothness**.) Consider an input to the Fourier head $x \in \mathbb{R}^n$, and denote by $f_x : [-1, 1] \to \mathbb{R}$ the optimal conditional distribution that we would like the Fourier head to approximate for this input. Suppose that there exists some $t \geq 2$ such that the Fourier coefficients of $f_x$ decay on the order of $1/k^t$. Denote by $f_{x,N}$ the truncation of $f_x$ to its first $N$ frequencies, denote by $\vec{b} \in \mathbb{R}^m$ the $m$ bin centerpoints in $[-1, 1]$, and denote by $y^{(N)} = f_{x,N}(\vec{b})/(f_{x,N}(b_0) + \cdots + f_{x,N}(b_{m-1})) \in \mathbb{R}^m$ the discretization of $f_{x,N}$ into $m$ bins. Then, there exist constants $C_1, C_2 > 0$ such that*

$$s(y^{(N)}) = C_1 - \frac{C_2}{N^{2t-1}} + O(1/N^{2t}). \tag{3.3}$$

*Proof of Claim 2 of Theorem 3.3.* We can estimate that

$$s(y^{(N)}) = \frac{1}{\sum_{j=0}^{m-1} f_{x,N}(b_j)} \sum_{\sigma=1}^{\infty} \alpha_\sigma \| f_{x,N}(\vec{b}) - g_\sigma * f_{x,N}(\vec{b}) \|_2 \qquad \text{(Definition 3.2)}$$

$$= \frac{1}{\sum_{j=0}^{m-1} f_{x,N}(b_j)} \sum_{\sigma=1}^{\infty} \alpha_\sigma \Big( \| B_1(m,\sigma) f_{x,N}(\vec{b}) - (\tilde{g}_\sigma * f_{x,N})(\vec{b}) \|_2 \qquad \text{(Lemma A.3)}$$
$$+ \sqrt{m} O(1/N^{2t+1}) \Big)$$

$$= \frac{\sqrt{m}}{\sum_{j=0}^{m-1} f_{x,N}(b_j)} \sum_{\sigma=1}^{\infty} \alpha_\sigma \Big( B_2(\sigma, m, x) - \frac{B_1(m,\sigma)^2 B_3(x)^2}{(2t-1)N^{2t-1}} \qquad \text{(Lemma A.5)}$$
$$+ O(1/N^{2t}) + O(1/N^{2t+1}) \Big)$$

$$= \frac{2}{\sqrt{m}} \cdot \Big( C_3 - \frac{C_4}{N^{2t-1}} + O(1/N^{2t}) \Big). \qquad \text{(Lemmata A.5, A.6)}$$

In the last step we used the convergence of the respective series (which follows from boundedness of $B_2(\sigma, m, x)$ and $B_1(m, \sigma)$ in $\sigma$) and we assigned $C_3$ and $C_4$ to be those sums. This completes the proof. $\qquad \square$

*Proof of Claim 1 of Theorem 3.3.* The proof of this claim is more straightforward. For any function $f$ on $[-1, 1]$ that is at least twice continuously differentiable, we know that the Fourier series of $f$

converges uniformly and absolutely to $f$ (Stein & Shakarchi, 2003, Ch. 2, Cor. 2.4). In other words, the function $f_N$ being learnt by the Fourier head converges uniformly and absolutely to $f$. $\qquad\square$

## A.4 PROVING THE LEMMATA

In this subsection, we will restate and prove Lemmata A.3, A.5, and A.6.

**Lemma A.3.** (Discrete convolution is close to continuous convolution) *If we define the constant* $B_1(m, \sigma) := 1 + \frac{G_\sigma(m)}{S(m,\sigma)}$, *then we have that*

$$\|f_{x,N}(\vec{b}) - g_\sigma * f_{x,N}(\vec{b})\|_2 = \|B_1(m,\sigma)f_{x,N}(\vec{b}) - \tilde{g}_\sigma * f_{x,N}(\vec{b})\|_2 + \sqrt{m}O(1/N^{2t+1}). \quad \text{(A.6)}$$

*Furthermore, $B_1(m, \sigma)$ satisfies the following bound, uniformly in $\sigma$,*

$$B_1(m, \sigma) \leq 1 + \frac{1}{2m - 1}. \quad \text{(A.7)}$$

*Proof of Lemma A.3.* Extending $f_{x,N}$ periodically to $[-2, 2]$, we can compute that the continuous convolution $\tilde{g}_\sigma * f_{x,N}(z)$ is

$$(\tilde{g}_\sigma * f_{x,N})(z) = \int_{-2}^{2} \tilde{g}_\sigma(u) f_{x,N}(z - u)\, du \quad \text{(A.13)}$$

$$= \frac{m}{2S(m,\sigma)} \int_{-2}^{2} G_\sigma\left(\frac{mu}{2}\right) f_{x,N}(z - u)\, du \quad \text{(A.14)}$$

$$= \frac{1}{S(m,\sigma)} \int_{-m}^{m} G_\sigma(s) f_{x,N}\left(z - \frac{2s}{m}\right) ds, \quad \text{(A.15)}$$

where in the third step we applied the change of variables $s = \frac{mu}{2}$. We claim that this is precisely a continuous approximation of the discrete convolution in Definition 3.2. To see this, we will apply the Euler-Maclaurin formula. This formula says that the integral in Equation A.15 is a Riemann sum over rectangles of width 1 evaluated at the right endpoints of each interval, minus an error term $E(m, \sigma)$, as follows:

$$(\tilde{g}_\sigma * f_{x,N})(b_j) + E(m, \sigma) = \frac{1}{S(m,\sigma)} \sum_{k=1-m}^{m} G_\sigma(k) \cdot f_{x,N}\left(b_j - \frac{2k}{m}\right) \quad \text{(A.16)}$$

$$= \frac{1}{S(m,\sigma)} \sum_{k=1-m}^{m} G_\sigma(k) \cdot f_{x,N}\left(-1 + \frac{2j+1}{m} - \frac{2k}{m}\right) \quad \text{(A.17)}$$

$$= \frac{1}{S(m,\sigma)} \sum_{k=1-m}^{m} G_\sigma(k) \cdot f_{x,N}\left(-1 + \frac{2(j-k)+1}{m}\right) \quad \text{(A.18)}$$

$$= \frac{1}{S(m,\sigma)} \left( \sum_{k=1-m}^{m-1} G_\sigma(k) \cdot f_{x,N}(b_{j-k}) + G_\sigma(m) f_{x,N}(b_{j-m}) \right) \quad \text{(A.19)}$$

$$= \frac{1}{S(m,\sigma)} \left( S(m,\sigma) \cdot (g_\sigma * f_{x,N})(b_j) + G_\sigma(m) f_{x,N}(b_j) \right) \quad \text{(A.20)}$$

$$= (g_\sigma * f_{x,N})(b_j) + \frac{1}{S(m,\sigma)} G_\sigma(m) f_{x,N}(b_j), \quad \text{(A.21)}$$

where the error term is defined as

$$E(m, \sigma) := \frac{1}{S(m,\sigma)} \int_{-m}^{m} \frac{d\left(G_\sigma(s) f_{x,N}\left(z - \frac{2s}{m}\right)\right)}{ds} P_1(s)\, ds \quad \text{(A.22)}$$

$$+ \frac{1}{2S(m,\sigma)} \left( G_\sigma(m) f_{x,N}(z - 2) - G_\sigma(-m) f_{x,N}(z + 2) \right), \quad \text{(A.23)}$$

where $P_1(s) := s - \lfloor s \rfloor - 1/2$ is the periodized Bernoulli polynomial. We will now estimate this error term. Note that since $G_\sigma$ is an even function and $f_{x,N}$ is periodic with period 2, the difference in A.23 is 0. Therefore, we can compute that

$$E(m, \sigma) = \frac{1}{S(m, \sigma)} \int_{-m}^{m} G'_\sigma(s) P_1(s) f_{x,N} \left( z - \frac{2s}{m} \right) ds \tag{A.24}$$

$$- \frac{2}{mS(m, \sigma)} \int_{-m}^{m} G_\sigma(s) P_1(s) f'_{x,N} \left( z - \frac{2s}{m} \right) ds. \tag{A.25}$$

Using the triangle inequality, we can bound $E(m, \sigma)$ in terms of convolutions with $\tilde{g}_\sigma$:

$$|E(m, \sigma)| \leq \frac{1}{S(m, \sigma)} \left| \int_{-m}^{m} G'_\sigma(s) P_1(s) f_{x,N} \left( z - \frac{2s}{m} \right) ds \right| \tag{A.26}$$

$$+ \frac{1}{S(m, \sigma)} \left| \int_{-m}^{m} G_\sigma(s) P_1(s) f'_{x,N} \left( z - \frac{2s}{m} \right) \frac{-2}{m} ds \right| \tag{A.27}$$

$$\leq \frac{1}{S(m, \sigma)} \left( \frac{m}{2\sigma^2} \int_{-m}^{m} G_\sigma(s) f_{x,N} \left( z - \frac{2s}{m} \right) ds + \right. \tag{A.28}$$

$$\left. + \frac{1}{m} \int_{-m}^{m} G_\sigma(s) \left| f'_{x,N} \left( z - \frac{2s}{m} \right) \right| ds \right) \tag{A.29}$$

$$= \frac{m}{2\sigma^2} (\tilde{g}_\sigma * f_{x,N})(z) + \frac{1}{m} (\tilde{g}_\sigma * |f'_{x,N}|)(z), \tag{A.30}$$

where in Equation A.29 we used that $|P_1(s)| \leq 1/2$ and that $|G'_\sigma(s)| = |s| G_\sigma(s)/\sigma^2 \leq mG_\sigma(s)/\sigma^2$ for $s \in [-m, m]$.

Note that since $\tilde{g}_\sigma$ is a truncated Gaussian on $[-2, 2]$, it is infinitely differentiable on the open set $(-2, 2)$, however, it is not differentiable at the endpoints $-2$ and $2$ when treated as a 4-periodic function. This technical difficulty can be resolved using mollifiers: we can replace $\tilde{g}_\sigma$ with $\tilde{g}_\sigma * \varphi_\epsilon$, where $\{\varphi_\epsilon\}$ is a family of mollifiers indexed by $\epsilon > 0$. The key properties of a mollifier are that $\tilde{g}_\sigma * \varphi_\epsilon$ is infinitely differentiable as a 4-periodic function for all $\epsilon > 0$ and $\lim_{\epsilon \to 0} \tilde{g}_\sigma * \varphi_\epsilon = \tilde{g}_\sigma$ (Stein & Shakarchi, 2005, Ch. 3). We are ultimately interested in only bounds on absolute values of $\tilde{g}_\sigma$ convolved with various functions, and since absolute values are continuous and inequalities are preserved under taking limits, all our bounds are still true. In particular, this shows that the $k$'th Fourier coefficients of $\tilde{g}_\sigma$ decay faster than any polynomial. And on the other hand, by assumption we know that the Fourier coefficients of $f_{x,N}$ decay on the order of $1/k^t$; and we know that $|f'_{x,N}|$ is continuous and $2\pi$ periodic, so its Fourier coefficients converge. So by the convolution theorem, we can deduce that the Fourier coefficients of $\tilde{g} * f_{x,N}$ and $\tilde{g}_\sigma * |f'_{x,N}|$ decay faster than any polynomial. Summed over the $N$ frequencies, this shows that $|\tilde{g}_\sigma * f_{x,N}(x)|$ and $|\tilde{g}_\sigma * |f'_{x,N}| (z)|$ decay faster than any polynomial as well. Since $m$ is fixed and $\sigma \geq 1$, this implies that

$$|E(m, \sigma)| = O(1/N^{2t+1}). \tag{A.31}$$

Using Definition A.1 and Equation A.16, we have that

$$g_\sigma * f_{x,N}(b_j) = \frac{1}{S(m, \sigma)} \sum_{k=1-m}^{m} G_\sigma(k) \cdot f_{x,N} \left( b_j - \frac{2k}{m} \right) - \frac{1}{S(m, \sigma)} G_\sigma(m) f_{x,N}(b_j) \tag{A.32}$$

$$= (\tilde{g}_\sigma * f_{x,N})(b_j) + E(m, \sigma) - \frac{1}{S(m, \sigma)} G_\sigma(m) f_{x,N}(b_j). \tag{A.33}$$

If we define $C_1(m, \sigma) := \frac{G_\sigma(m)}{S(m, \sigma)}$, then Equation A.33 combined with A.31 together imply that

$$\left| g_\sigma * f_{x,N}(b_j) - \tilde{g}_\sigma * f_{x,N}(b_j) + C_1(\sigma, m) f_{x,N}(b_j) \right| = O(1/N^{2t+1}). \tag{A.34}$$

Finally, we can estimate that

$$\|f_{x,N}(\vec{b}) - g_\sigma * f_{x,N}(\vec{b})\|_2 \tag{A.35}$$

$$= \|f_{x,N}(\vec{b}) - \tilde{g}_\sigma * f_{x,N}(\vec{b}) + C_1(\sigma, m)f_{x,N}(\vec{b})\|_2 \tag{A.36}$$

$$+ \|g_\sigma * f_{x,N}(\vec{b}) - \tilde{g}_\sigma * f_{x,N}(\vec{b}) + C_1(m, \sigma)f_{x,N}(\vec{b})\| \tag{A.37}$$

$$= \|f_{x,N}(\vec{b}) - \tilde{g}_\sigma * f_{x,N}(\vec{b}) + C_1(m, \sigma)f_{x,N}(\vec{b})\|_2 + \sqrt{m}\|O(1/N^{2t+1})\|_2 \tag{A.38}$$

$$= \|(1 + C_1(m, \sigma))f_{x,N}(\vec{b}) - \tilde{g}_\sigma * f_{x,N}(\vec{b})\|_2 + \sqrt{m}O(1/N^{2t+1}). \tag{A.39}$$

This completes the first part of the proof. For the second part of the proof, since $S(m, \sigma) \geq (2m - 1)G_\sigma(m - 1)$, we can estimate that

$$B_1(m, \sigma) \leq 1 + \frac{G_\sigma(m)}{(2m-1)G_\sigma(m-1)} \leq 1 + \frac{1}{2m-1}e^{-(2m-1)/2\sigma^2} \leq 1 + \frac{1}{2m-1}. \tag{A.40}$$

This completes the proof. $\qquad\square$

**Lemma A.5.** (Main term asymptotic) *Denote by $a_0(x)$ the constant coefficient of $f_{x,N}$. Let us suppose that the Fourier coefficients of $f_{x,N}$ decay like $B_3(x)/k^t$, and define the constant*

$$B_2(\sigma, m, x) := \sqrt{a_0(x)^2 B_1(m, \sigma)^2 + 2B_1(m, \sigma)^2 B_3(x)^2 \zeta(2t)}. \tag{A.9}$$

*Then we know that*

$$\|B_1(m, \sigma)f_{x,N}(\vec{b}) - \tilde{g}_\sigma * f_{x,N}(\vec{b})\|_2 \tag{A.10}$$

$$= \sqrt{m}\left(B_2(\sigma, m, x) - \frac{B_1(m, \sigma)^2 B_3(x)^2}{2t-1} \cdot \frac{1}{N^{2t-1}} + O(1/N^{2t})\right). \tag{A.11}$$

*Furthermore, $B_2(\sigma, m, x)$ is bounded from above and below as a function of $\sigma$ and $m$.*

*Proof of Lemma A.5.* We will first argue that $B_2(\sigma, m, x)$, as a function of $\sigma$ and $m$, is bounded from above and below. Indeed, from its definition, $B_2(\sigma, m, x) \geq \sqrt{a_0(x)^2 B_1(m, \sigma)^2} \geq |a_0(x)|$. But the Fourier PDF has integral 1 over $[-1, 1]$, so its constant term is $a_0(x) = 1/2$. This implies that $B_2(\sigma, m, x) \geq 1/2$. To see $B_2(\sigma, m, x)$ is bounded from above, we simply recall that in Lemma A.3 we showed that $|B_1(m, \sigma)| \leq 2$, which implies that $|B_2(\sigma, m, x)| \leq \sqrt{4a_0(x)^2 + 8B_3(x)^2\zeta(2t)}$. This shows that $B_2(\sigma, m, x)$ is bounded above and below as a function of $m$ and $\sigma$, as claimed.

Now, let $(d_0(x), \ldots, d_{m-1}(x)) \in \mathbb{R}^m$ be the discrete Fourier transform of $(B_1(m, \sigma)f_{x,N} - \tilde{g}_\sigma * f_{x,N})(\vec{b}) \in \mathbb{R}^m$. For notational simplicity, we will write $B_1 = B_1(m, \sigma)$ as long as $\sigma$ is fixed. By Plancherel's Theorem, we have

$$\sum_{j=0}^{m-1} |(B_1 f_{x,N} - \tilde{g}_\sigma * f_{x,N})(b_j)|^2 = \frac{1}{m}\sum_{k=0}^{m-1} |d_k(x)|^2. \tag{A.41}$$

Let $h_{\sigma,j}$ be the Fourier coefficients of $\tilde{g}_\sigma$, treated as a periodic function with period 4, and defined over $[-2, 2]$:

$$\tilde{g}_\sigma(z) = \sum_{j=-\infty}^{\infty} h_{\sigma,j}e^{\pi ijz/2}. \tag{A.42}$$

Since $f_{x,N}$ is defined over $[-1, 1]$ and is periodic with period 2, we can likewise treat it as a function over $[-2, 2]$ with period 4, in which case we can rewrite its Fourier series as

$$f_{x,N}(z) = \sum_{j=-2N}^{2N} \tilde{a}_j(x)e^{\pi ijz/2}, \tag{A.43}$$

where

$$\tilde{a}_j(x) = \begin{cases} a_{j/2}(x) & \text{if } j \equiv 0 \pmod 2 \\ 0 & \text{else.} \end{cases} \tag{A.44}$$

Then, by the Convolution Theorem, we have

$$(B_1 f_{x,N} - \tilde{g}_\sigma * f_{x,N})(b_l) = \sum_{j=-2N}^{2N} \tilde{a}_j(x)(B_1 - h_{\sigma,j}) \cdot e^{\pi i j b_l/2} \tag{A.45}$$

$$= \sum_{k=-N}^{N} \tilde{a}_{2k}(x)(B_1 - h_{\sigma,2k}) \cdot e^{\pi i k b_l} \tag{A.46}$$

$$= \sum_{k=-N}^{N} a_k(x)(B_1 - h_{\sigma,2k}) \cdot e^{\pi i k b_l}, \tag{A.47}$$

where in the second equality we used the fact that $\tilde{a}_j$ is 0 for odd $j$ and therefore re-indexed using $k = j/2$. Thus, using the definition of DFT along with Equation A.47, we get

$$d_k(x) = \sum_{l=0}^{m-1} (B_1 f_{x,N} - \tilde{g}_\sigma * f_{x,N})(b_l) \cdot e^{-2\pi i k l/m} \tag{A.48}$$

$$= \sum_{l=0}^{m-1} \sum_{j=-N}^{N} a_j(x)(B_1 - h_{\sigma,2j})e^{\pi i j b_l} \cdot e^{-2\pi i k l/m} \tag{A.49}$$

$$= \sum_{j=-N}^{N} a_j(x)(B_1 - h_{\sigma,2j}) \sum_{l=0}^{m-1} e^{\pi i j(-1+\frac{2l+1}{m})} \cdot e^{-2\pi i k l/m} \tag{A.50}$$

$$= \sum_{j=-N}^{N} a_j(x)(B_1 - h_{\sigma,2j}) \cdot e^{\pi i j(-1+1/m)} \sum_{l=0}^{m-1} e^{2\pi i (j-k)l/m}. \tag{A.51}$$

First, we claim that at most a single summand (in $j$) is represented. Towards this, we note that

$$\sum_{l=0}^{m-1} e^{2\pi i (j-k)l/m} = \begin{cases} 0 & j \not\equiv k \pmod{m} \\ m & j \equiv k \pmod{m}. \end{cases} \tag{A.52}$$

Then, we note that since $0 < N < m/2$, for each $0 \le k < m$, there is at most one $j \in \{-N, -N+1, \ldots, N\}$, such that $j \equiv k \pmod{m}$. This shows that there is at most a single summand. We will now find the exact formula for each summand. We consider three cases.

- Case 1: $0 \le k \le N$. In this case, $j = k$ satisfies $j \equiv k \pmod{m}$, so this index gives the exponential sum of $m$.

- Case 2: $N < k < m - N$. In this case, $k$ is too large to be an index in the sum, so we can't choose $j = k$; the next smallest equivalent value is $j = k - m$, which satisfies $j \equiv k \pmod{m}$. But $N - m < j < -N$ in this case, so $k$ is too small to be an index in the sum; therefore, every exponential sum is zero in this range.

- Case 3: $m - N \le k \le m - 1$. In this case, $j = k - m$ satisfies $j \equiv k \pmod{m}$. We have $-N \le j \le 1$, so this is a valid index in the sum.

This gives the following closed formula:

$$d_k(x) = \begin{cases} m \cdot a_k(x)(B_1 - h_{\sigma,2k})e^{\pi i k(-1+1/m)} & 0 \le k \le N \\ 0 & N < k < m - N \\ m \cdot a_{k-m}(B_1 - h_{\sigma,2(k-m)})e^{\pi i(k-m)(-1+1/m)} & m - N \le k \le m - 1. \end{cases} \tag{A.53}$$

Using this closed formula in A.41, we obtain

$$\sum_{j=0}^{m-1} \left| (B_1 f_{x,N} - \tilde{g}_\sigma * f_{x,N})(b_j) \right|^2 \tag{A.54}$$

$$= \frac{1}{m} \sum_{k=0}^{N} \left| m \cdot a_k(x)(B_1 - h_{\sigma,2k}) e^{\pi i k(-1+1/m)} \right|^2 \tag{A.55}$$

$$+ \frac{1}{m} \sum_{k=m-N}^{m-1} \left| m \cdot a_{k-m}(B_1 - h_{\sigma,2(k-m)}) e^{\pi i (k-m)(-1+1/m)} \right|^2 \tag{A.56}$$

$$= m \sum_{k=0}^{N} \left| a_k(x)(B_1 - h_{\sigma,2k}) \right|^2 + m \sum_{k=m-N}^{m-1} \left| a_{k-m}(B_1 - h_{\sigma,2(k-m)}) \right|^2, \tag{A.57}$$

where in the last step we used that $\left| e^{\pi i k(1-1/m)} \right| = 1 = \left| e^{\pi i (k-m)(1-1/m)} \right|$ since they are both complex exponentials. Now, since $\tilde{g}_\sigma$ is a real and even function, we know that its Fourier coefficients $h_{\sigma,k}$ are real. Further, since $\tilde{g}_\sigma$ is infinitely differentiable, we also know that $h_{\sigma,k} = O(1/k^t)$ (in fact they decay faster than $1/k^p$ for any $p \geq 1$). Thus, using that $|a_k(x)|$ decays like $B_3(x)/k^t$, we see

$$|a_k(x)(B_1 - h_{\sigma,2k})|^2 = |a_k(x)|^2 \left( B_1^2 - 2 h_{\sigma,2k}(x) + h_{\sigma,2k}^2 \right) \tag{A.58}$$

$$= \frac{B_3^2 B_1^2}{k^{2t}} + O\left( \frac{1}{k^{2t}(2k)^t} \right) + O\left( \frac{1}{k^{2t}(2k)^{2t}} \right). \tag{A.59}$$

From A.59, it is clear that since we are interested in only the dominant asymptotic, we can safely ignore the higher order terms coming from the $h_{\sigma,k}$. As a result, we can estimate that

$$\sum_{j=0}^{m-1} \left| (B_1 f_{x,N} - \tilde{g}_\sigma * f_{x,N})(b_j) \right|^2 \approx m B_1^2 a_0(x)^2 + m \sum_{k=1}^{N} \frac{B_3^2 B_1^2}{k^{2t}} + m \sum_{k=m-N}^{m-1} \frac{B_3^2 B_1^2}{(k-m)^{2t}} \tag{A.60}$$

$$= m a_0(x)^2 B_1^2 + m \sum_{k=1}^{N} \frac{B_3^2 B_1^2}{k^{2t}} + m \sum_{k=-N}^{-1} \frac{B_3^2 B_1^2}{k^{2t}} \tag{A.61}$$

$$= m a_0(x)^2 B_1^2 + m \sum_{k=1}^{N} \frac{B_3^2 B_1^2}{k^{2t}} + m \sum_{k=1}^{N} \frac{B_3^2 B_1^2}{k^{2t}} \tag{A.62}$$

$$= m a_0(x)^2 B_1^2 + 2m \sum_{k=1}^{N} \frac{B_3^2 B_1^2}{k^{2t}}. \tag{A.63}$$

Next, we note that our asymptotic in Lemma A.4, applied at $2t$, yields

$$\sum_{k=1}^{N} \frac{1}{k^{2t}} = \zeta(2t) - \frac{1}{2t-1} \frac{1}{N^{2t-1}} + O(1/N^{2t}). \tag{A.64}$$

Substituting this into A.63, we obtain

$$\sum_{j=0}^{m-1} \left| (B_1 f_{x,N} - \tilde{g}_\sigma * f_{x,N})(b_j) \right|^2 \tag{A.65}$$

$$= m a_0(x)^2 B_1^2 + 2m B_1^2 B_3^2 \left( \zeta(2t) - \frac{1}{2t-1} \frac{1}{N^{2t-1}} + O(1/N^{2t}) \right) \tag{A.66}$$

$$= m \left( B_2^2 - \frac{2 B_1^2 B_3^2}{2t-1} \frac{1}{N^{2t-1}} + B_1^2 B_3^2 O(1/N^{2t}) \right) \tag{A.67}$$

$$= m \left( B_2^2 - \frac{2 B_1^2 B_3^2}{2t-1} \frac{1}{N^{2t-1}} + O(1/N^{2t}) \right), \tag{A.68}$$

where we defined $B_2 = B_2(\sigma, m, x) := \sqrt{a_0(x)^2 B_1^2 + 2B_1^2 B_3^2 \zeta(2t)}$ as in the statement of the Lemma, and in the third line we applied Lemma A.3 to estimate that $B_1 \leq 2$, and we used that $B_3 = B_3(x)$ only depends on $x$. Then, using the Taylor expansion $(1 + x)^{1/2} = 1 + \frac{x}{2} + O(x^2)$ about 0, we can estimate that

$$\left( \sum_{j=0}^{m-1} |(B_1 f_{x,N} - \tilde{g}_\sigma * f_{x,N})(b_j)|^2 \right)^{1/2} \tag{A.69}$$

$$= \left( mB_2^2 - \frac{m 2 B_1^2 B_3^2}{2t-1} \frac{1}{N^{2t-1}} + mO(1/N^{2t}) \right)^{1/2} \tag{A.70}$$

$$= \sqrt{m} B_2 \left( 1 - \frac{2B_1^2 B_3^2}{(2t-1)B_2^2} \frac{1}{N^{2t-1}} + \frac{1}{B_2^2} O(1/N^{2t}) \right)^{1/2} \tag{A.71}$$

$$= \sqrt{m} B_2 \left( 1 - \frac{1}{2} \cdot \frac{2B_1^2 B_3^2}{(2t-1)B_2^2} \frac{1}{N^{2t-1}} + O(1/N^{2t}) \right) \tag{A.72}$$

$$= \sqrt{m} \left( B_2 - \frac{B_1^2 B_3^2}{2t-1} \cdot \frac{1}{N^{2t-1}} + O(1/N^{2t}) \right). \tag{A.73}$$

To justify our application of the Taylor expansion, we note that $N \gg 1$, and $B_2 = B_2(\sigma, m, x)$ is bounded below as a function of $\sigma$ and $m$. This completes the proof. □

**Lemma A.6.** (The average value of the truncated Fourier PDF is $1/2$) *If $N < m/2$, then*

$$\sum_{j=0}^{m-1} f_{x,N}(b_j) = \frac{m}{2}. \tag{A.12}$$

*Proof of Lemma A.6.* Denote by $a_k$ the Fourier coefficients of $f_{x,N}$. We can compute that

$$\sum_{j=0}^{m-1} f_{x,N}(b_j) = \sum_{j=0}^{m-1} \sum_{k=-N}^{N} a_k e^{i\pi k \left( -1 + \frac{2j+1}{m} \right)} \tag{A.74}$$

$$= \sum_{k=-N}^{N} a_k e^{-i\pi k} e^{\frac{i\pi k}{m}} \sum_{j=0}^{m-1} e^{\frac{2\pi ijk}{m}}. \tag{A.75}$$

Note that by hypothesis, $N < m/2$, which implies that $|k| < m$ for every outer sum index $k$. We consider two cases; if $k = 0$, then the innermost summand is $e^{\frac{2\pi ijk}{m}} = 1$; and if $k \neq 0$, then the innermost sum is a truncated geometric series with first term 1, common ratio $e^{\frac{2\pi ik}{m}}$, and $m$ terms. In summary, the innermost summand is

$$\sum_{j=0}^{m-1} e^{\frac{2\pi ijk}{m}} = \begin{cases} m & k = 0 \\ 0 & k \neq 0, \end{cases} \tag{A.76}$$

which implies that $\sum_{j=0}^{m-1} f_{x,N}(b_j) = ma_0$. But $a_0 = 1/2$ because $f_{x,N}$ is a PDF implies it has average value $1/2$ over $[-1,1]$. This completes the proof. □

# B    SMOOTHNESS METRIC

We will examine how the proposed smoothness metric Equation 3.1 behaves in a toy example setting to gain intuition for its behavior. Consider a square wave, which can be expressed as an infinite sum of odd integer harmonics that decay in amplitude proportional to their frequency:

$$f(x) = \frac{4}{\pi} \sum_{n=1,3,5,\dots}^{\infty} \frac{1}{n} \sin\left( \frac{n\pi x}{L} \right). \tag{B.1}$$

Here, the wavelength is $2L$ (Weisstein, 2024).

We construct a truncated version of the square wave with a finite and fixed number of frequencies. The waveform will slowly approach its jagged, square shape as more sine waves are added. We frame these increasingly jagged waves as discretized multinomial densities to simulate the output of the Fourier head. To do this, we simply set the height to zero when the wave crest becomes negative and normalize the sum to 1. The output of this transformation for a few representative waveforms is pictured in Figure 5.

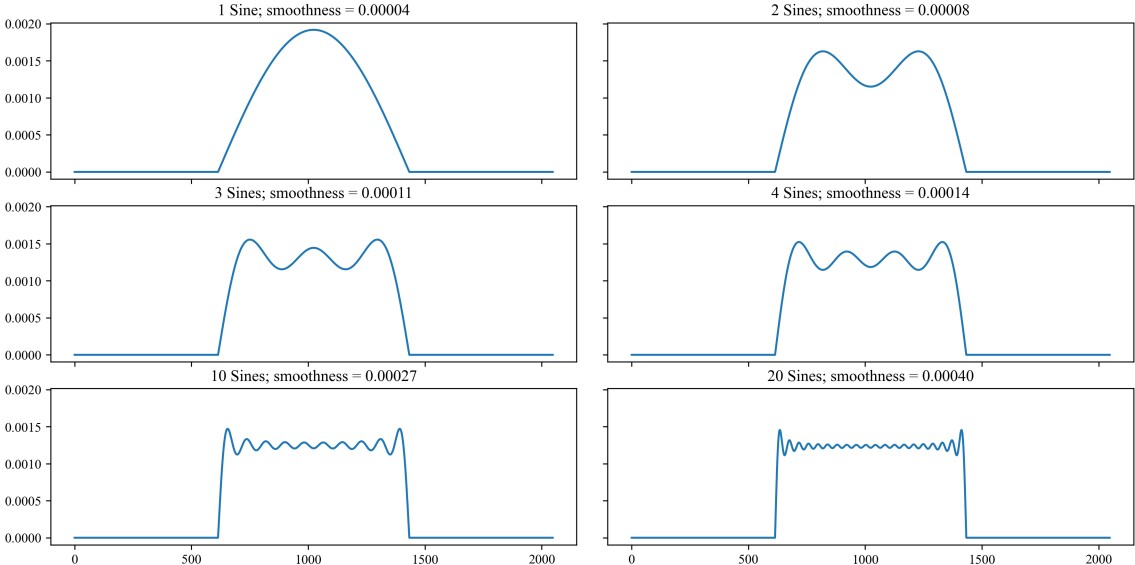

Figure 5: Truncated square waves framed as densities and their smoothness.

Intuitively, the truncated square wave with a single sine wave ought to be the smoothest. Thus our metric in this context should be smallest at that point, and increase monotonically as we add more sine waves. The plot in 6 demonstrates that this is indeed the case.

**Choice of L2 Distance over L1 Distance:** The proposed smoothness metric Equation 3.1 permits a general measure of discrepancy $D$, and we've chosen $D$ to be $L^2$ distance as indicated in 3.2. We empirically observe that $L^2$ distance better preserves monotonicity than the $L^1$ for higher frequency content, thus motivating this choice. With a sample rate of $2048$Hz, the $L^1$ distance exhibits some undesirable warping when our square-wave multinomial uses over 80 sine waves (see Figure 6). A Fourier head in a practical setting may possess several more than $80$ frequencies; accordingly, we favor the $L^2$ distance as our discrepancy measure.

**Alternative Notions of Smoothness:** In validating our choice of smoothness metric, we compare it to the *spectral entropy* (Inouye et al., 1991), which has a similar purpose in quantifying the "smoothness" of the frequency content of a signal. Spectral entropy is defined as the Shannon entropy of the *power spectral density* of a sampled signal $f$, which is defined as follows:

$$H(f; N) = \sum_{n \in N} p(n) \log_2 \left( \frac{1}{p(n)} \right) = -\sum_{n \in N} \frac{S_n}{S_{total}} \log_2 \left( \frac{S_n}{S_{total}} \right) \tag{B.2}$$

Here, $N$ is the number of Fourier frequencies and $S$ is the power of a frequency $n \in N$; $S_n$ is the power spectrum of the $n$th frequency, and $S_{total}$ is the power of the signal using all $N$ frequencies. For some frequency at index $n$, $S_n/S_{total}$ is called its relative power and $\sum_{n \in N} \frac{S_n}{S_{total}} = 1$ enables us to consider each frequency's power as a probability.

In the discrete case, the maximum entropy distribution is the uniform distribution. Thus, white noise will have the highest spectral entropy. This has the consequence that power spectral densities have more high frequency information will have lower entropy than that of white noise, provided that there is a relationship between amplitude and frequency. More concretely, blue noise, which is

Smoothness Metric on Square Wave with increasing Number of Sine Waves

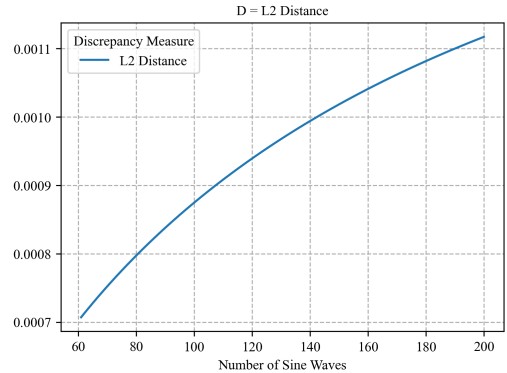 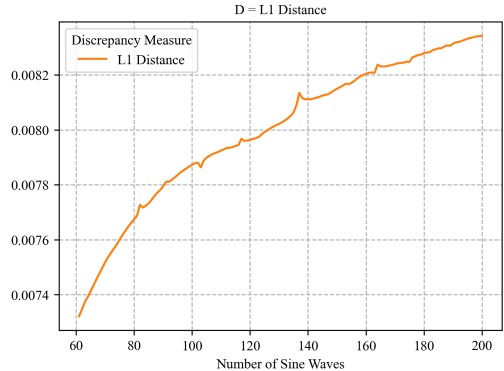

Figure 6: Values of the smoothness metric 3.2 on our square-wave-like multinomials as we increase the number of sine waves. We desire the value of this metric to be close to zero when there are few sine waves, and be monotonically increasing with each additional wave, indicating that adding more high frequency content results in a less smooth distribution. On the right, we can see that $L^1$ as a discrepancy measure leads to non-monotonicity, motivating our choice of $L^2$ distance in measuring our results.

defined by the amplitude increasing proportionally to the frequency, will have lower spectral entropy than white noise. We sought a metric that always quantified 'sharper' signals like blue noise as less smooth. In Table 4, we frame sampled noises of different types as multinomial distributions to match our model setting by normalizing their amplitudes to be in $[0, 1]$ and normalizing their sum to 1. Our noise types are defined before normalization, in order of smoothest to sharpest:

- Brown: $S \propto \frac{1}{F^2}$

- Pink: $S \propto \frac{1}{F}$

- White: $S \sim \mathcal{N}(0, 1)$

- Blue: $S \propto F$

where $S$ is the power density and $F$ is the frequency. To obtain samples of each type, we first generate white noise. We do this by sampling a Gaussian with mean 0 and standard deviation 1 to obtain amplitudes for $t$ samples. We then apply the Fourier transform, and multiply (or divide) the amplitudes of each component by their frequency, and apply the inverse Fourier transform to recover the waveform. Finally we adjust the range of amplitudes of the signal to be within $[0, 1]$ and normalize the sum to 1.

| Discrepancy | Noise | Mean ± Std. Deviation | Diff | Delta | Desired Delta |
|---|---|---|---|---|---|
| L2 | Brown | $0.0003 \pm 0.0001$ | n/a | n/a | n/a |
| L2 | Pink | $0.0017 \pm 0.0002$ | 0.0014 | + | + |
| L2 | White | $0.0034 \pm 0.0003$ | 0.0016 | + | + |
| L2 | Blue | $0.0038 \pm 0.0003$ | 0.0005 | + | + |
| Spectral Entropy | Brown | $0.4516 \pm 0.0894$ | n/a | n/a | n/a |
| Spectral Entropy | Pink | $0.3878 \pm 0.0603$ | -0.0638 | - | + |
| Spectral Entropy | White | $0.4266 \pm 0.0614$ | 0.0388 | + | + |
| Spectral Entropy | Blue | $0.4191 \pm 0.0583$ | -0.0076 | - | + |

Table 4: Smoothness measurements for four types of noise bootstrap aggregated over 1,000 trials. The color red emphasizes how the value of Spectral Entropy is undesirably not monotonic increasing for what we consider increasingly "sharp" noise types.

## C ADDITIONAL EXPERIMENT DETAILS, TOY EXAMPLES

### C.1 MOTIVATING EXAMPLE: AUDIO SPECTROGRAM TRANSFORMER

To illustrate a simple problem setting where the design of the Fourier head is appropriate, we use it as a drop-in replacement for a linear classification head in the Audio Spectrogram Transformer (Gong et al., 2021). We consider the task of beats per minute (BPM) classification for metronome-like audio samples (Wei et al., 2024) within the tempo range $\{50, 51, \ldots, 210\}$. While this task is not difficult, we use this audio classification task to illustrate some of the design choices one can make when using the Fourier head. In this case, it is natural to group the BPMs into contiguous bins $\{[50, 54], [55, 59], \ldots\}$ and use the Fourier head to classify them. These bins have a natural continuous structure, which is where the Fourier head performs well. We also expect that the categorical distribution over possible BPMs for a given audio clip ought to be unimodal and therefore require few frequencies to approximate. In fact, our best performing model for this example uses only one frequency.

We initialize the Audio Spectrogram Transformer with pretrained weights from AudioSet (Gemmeke et al., 2017), and we train two different models–one with a standard linear classification head, and one with the Fourier head. The Fourier head outperforms the linear classification head by an F1 score improvement of $+118\%$. We attribute this success to the inductive bias of continuity that the Fourier head imparts. In Figure 7 we present the learned probability masses of both heads on the same input sample. This graph illustrates that the Fourier head learns smoother PMFs than the linear head, a concept which we will later formalize and explore.

Audio Classification Task: Learned Linear vs. Fourier PMFs

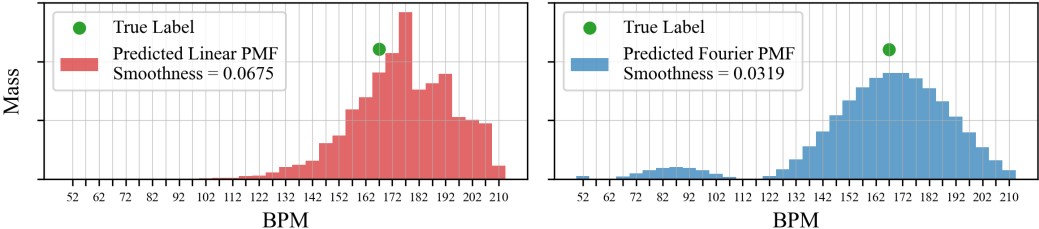

Figure 7: Comparison between the PMF learned by the linear head, and the Fourier head with 2 frequencies, for the toy BPM classification task, on a single audio example. We observe that the Fourier head learns a smoother categorical distribution over its predicted values, and is better centered around the ground truth label. We also note the small mini-sine wave artifacting on the left side of the Fourier model, which tends to occur when using few frequencies.

### C.2 LEARNING A CONTINUOUS DENSITY

Here we provide full details of the datasets used in our toy example of learning a known conditional distribution.

**Dataset**: We create a synthetic dataset $\mathcal{D} = \{(q(x), q(y), q(z))\} \subset \mathbb{R}^3$ as follows. Fix a probability distribution $\mathcal{P}_1 = \mathcal{P}_1(x)$ that is parameterized by one variable and a second distribution $\mathcal{P}_2 = \mathcal{P}_2(x, y)$ parameterized by two variables. Fix an interval $I \subset \mathbb{R}$. Sample $x$ uniformly from $I$, sample $y \sim \mathcal{P}_1(x)$, and finally sample $z \sim \mathcal{P}_2(x, y)$. We can repeat this sampling procedure $N$ times to obtain a set of $N$ triples for which we know the conditional distribution of $z$ given $x$ and $y$. Finally, we quantize this set to a fixed number of uniformly spaced bins in the range $[-1, 1]$ to obtain the dataset $\mathcal{D}_{\mathcal{P}_1, \mathcal{P}_2}$. We will denote the quantization of $z$ by $q(z)$. We quantize into 50 bins and our dataset has size 5000, with a 80-20 split between the train and test set. We describe three choices for the distributions we used to create our datasets. We fix $I = [-0.8, 0.8]$ and $\sigma^2 = 0.01$ in all of them.

1. *Gaussian dataset:* $\mathcal{P}_1(x) = \mathcal{N}(x, \sigma^2)$, and $\mathcal{P}_2(x, y) = \mathcal{N}(y, \sigma^2)$.
2. *GMM-2 dataset:* $\mathcal{P}_1 = \text{Uniform}(I)$, and $\mathcal{P}_2(x, y)$ is a GMM centered at $x$ and $y$ with variance $\sigma^2$.

3. *Beta dataset*: $\mathcal{P}_1(x) = \mathcal{N}(x, \sigma^2)$, and $\mathcal{P}_2(x, y) \sim U(\{\pm 1\}) \times \text{Beta}(100 |x|, 100 |y|)$, where $U(\{\pm 1\})$ denotes the Rademacher distribution supported on $\{\pm 1\}$ with probability $1/2$ each.

**Additional results:** In Figure 8, we present results from training over a range of frequencies, and for each frequency we ran experiments with and without Fourier regularization. In Table 6 we present results on the MSE metric, that show that the Fourier head outperforms the linear classification head.

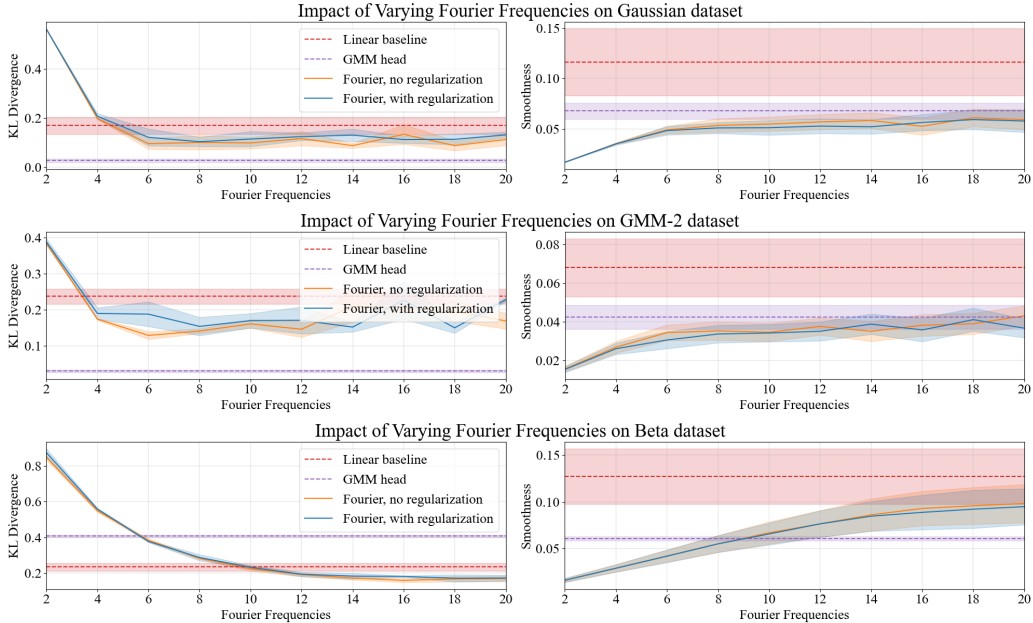

Figure 8: We study how the quantity of Fourier frequencies impacts KL divergence and smoothness for the toy example on each dataset. For both KL divergence and smoothness, lower is better. We observe that the Fourier models with and without regularization performed similarly to each other, and outperformed the linear baseline. We also note that the 50% error bars are larger for the linear baseline model; this indicates that the Fourier models (both with and without regularization) are in general more stable. This is in contrast to our large scale time series forecasting experiments, where we find that regularization helps; this is likely because those experiments use an order of magnitude more frequencies, and their conditional distributions are more complicated. While the GMM head has better KL divergence on the Gaussian and GMM-2 datasets, which is to be expected, the Fourier model (both with and without regularization) eventually has the best KL divergence on the Beta dataset, since it is non-Gaussian Notice also how on each of the datasets, the smoothness degrades as frequency increases, in a fashion that follows the asymptotic from our Theorem 3.3.

### C.3 MLE-BASED FOURIER HEAD

We carry out experiments in the continuous domain analogous to those we did in the quantized domain from the toy example.

**Dataset**: We use the same synthetic datasets–Gaussian, GMM-2, and Beta–as in the previous subsection, except we do not quantize the data into bins.

**Task**: Predict the conditional distribution of $z$ given $(x, y)$.

**Model architecture**: Our model is an MLP with one hidden layer and the final layer is an MLE-based Fourier head which returns the $2N + 1$ learned real coefficients of the Fourier series, mapping $\mathbb{R}^2 \to \mathbb{R}^{64} \to \mathbb{R}^{32} \to \mathbb{R}^{2N+1}$. Alongside the Fourier-MLE model, we consider a baseline where the final layer is a Gaussian model mixture whose means and standard deviations are learned using an MLE objective. For the MLE-Fourier model, we sweep over frequencies $N = 2, 4, \ldots, 20$ and regularization $\gamma \in \{0, 10^{-6}\}$.

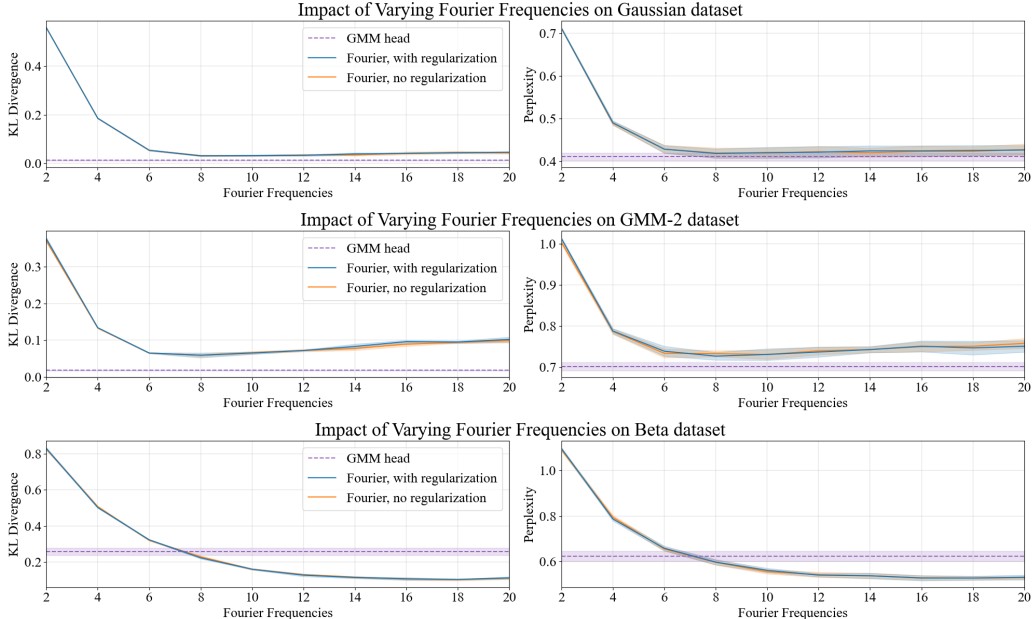

Figure 9: We study how the quantity of Fourier frequencies impacts KL divergence and perplexity for the toy example on each dataset for the MLE experiments. For both KL divergence and perplexity, lower is better. We observe that the Fourier models with and without regularization performed similarly to each other. While the GMM head has better KL divergence on the Gaussian and GMM-2 datasets, which is to be expected, the Fourier model (both with and without regularization) has the best KL divergence on the Beta dataset for sufficiently large Fourier frequencies, since it is non-Gaussian.

| | KL Divergence ($\downarrow$) | | |
|---|---|---|---|
| Dataset | Linear | GMM | Fourier |
| Gaussian | $0.170 \pm 0.052$ | $\mathbf{0.026} \pm 0.011$ | $0.116 \pm 0.043$ |
| GMM-2 | $0.238 \pm 0.032$ | $\mathbf{0.030} \pm 0.006$ | $0.146 \pm 0.033$ |
| Beta | $0.234 \pm 0.032$ | $0.407 \pm 0.012$ | $\mathbf{0.191} \pm 0.016$ |

| | Smoothness ($\downarrow$) | | |
|---|---|---|---|
| Dataset | Linear | GMM | Fourier |
| Gaussian | $0.116 \pm 0.049$ | $0.068 \pm 0.012$ | $\mathbf{0.057} \pm 0.011$ |
| GMM-2 | $0.068 \pm 0.022$ | $0.043 \pm 0.009$ | $\mathbf{0.038} \pm 0.007$ |
| Beta | $0.127 \pm 0.044$ | $\mathbf{0.061} \pm 0.003$ | $0.076 \pm 0.021$ |

Table 5: KL divergence and Smoothness for the three classification heads (Linear, GMM, and Fourier) on each of the three synthetic datasets (Gaussian, GMM-2, Beta). As expected, the GMM head achieves the best KL divergence on the Gaussian and GMM-2 datasets, as their conditional distributions are Gaussian. However, the Fourier head has the best KL divergence on the Beta dataset. This demonstrates the flexibility of the Fourier head in modeling non-Gaussian distributions as well.

**Model evaluation**: We use two metrics for evaluation. The first metric is the average KL divergence $D_{\mathrm{KL}}(\mathcal{P}(x,y)\|M(x,y))$, where $\mathcal{P}(x,y)$ is the fixed conditional distribution of $z$ given $(x,y)$ and $M(x,y)$ denotes the predicted probability density function of $z$. Our second metric is perplexity, which is the exponential of the average negative log likelihood of the test set.

**Results**: The metrics for the best performing model on each dataset are reported in Table 7. Figure 11 presents sample visualizations of the learned conditional distributions alongside the true densities. While, as expected, the GMM-MLE head outperforms the Fourier-MLE head on the Gaussian and GMM-2 datasets due to the Gaussian nature of the datasets, the Fourier-MLE head outperforms

Toy Example: MSE ($\downarrow$)

| Dataset | Pointwise Regression | Classification Head | | |
|---|---|---|---|---|
| | | Linear | GMM | Fourier |
| Gaussian | $0.010 \pm 0.001$ | $0.013 \pm 0.001$ | $0.010 \pm 0.001$ | $0.012 \pm 0.001$ |
| GMM-2 | $0.121 \pm 0.004$ | $0.126 \pm 0.004$ | $0.120 \pm 0.004$ | $0.123 \pm 0.005$ |
| Beta | $0.275 \pm 0.009$ | $0.276 \pm 0.008$ | $0.273 \pm 0.009$ | $0.275 \pm 0.008$ |

Table 6: We compare the MSE between the linear head, GMM head, and the Fourier head with 12 frequencies and no regularization, for every dataset in the toy example. We also include a Pointwise Regression model baseline, whose base architecture is same as the classification heads, except the last classification layer is replaced with a dense layer having output dimension 1. We train the Pointwise Regression model using MSE. For a given dataset, the MSE values across all of the models is roughly similar. This is because the pointwise regression model tends to regress to the mean, as does the expected value of each of the classification heads.

Toy Example: Ground Truth Conditional Distribution vs. Pointwise Regression Output

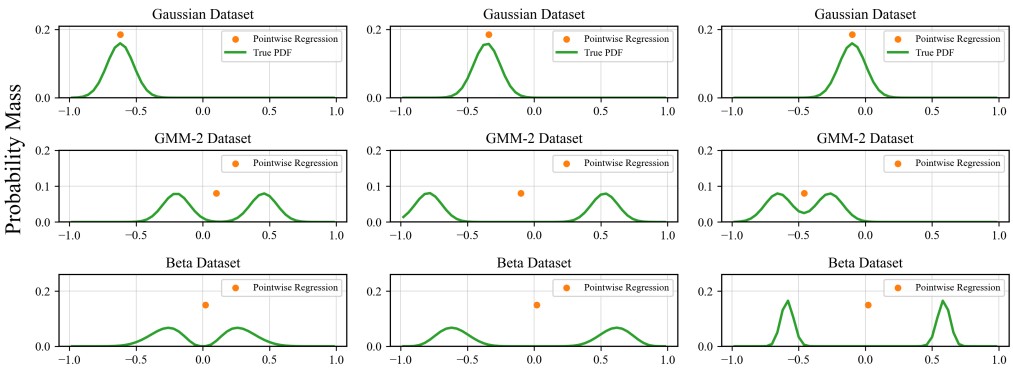

Figure 10: We present some examples of the ground truth conditional distribution versus the point predicted by the Pointwise Regression model. The regression model simply regresses to the mean of the conditional distribution. Accordingly, the regression model performs extremely well for the unimodal Gaussian dataset, and it performs poorly for the bimodal datasets GMM-2 and Beta.

the GMM-MLE head on the Beta dataset, highlighting the flexibility of the Fourier-MLE head in learning a large variety of probability distributions. In Appendix C.2, we present the results of a study on the impact of number of frequencies and Fourier regularization in the MLE setting.

| Dataset | KL Divergence ($\downarrow$) | | Perplexity ($\downarrow$) | |
|---|---|---|---|---|
| | GMM-MLE | Fourier-MLE | GMM-MLE | Fourier-MLE |
| Gaussian | $\mathbf{0.012} \pm 0.002$ | $0.034 \pm 0.003$ | $\mathbf{0.410} \pm 0.014$ | $0.422 \pm 0.019$ |
| GMM-2 | $\mathbf{0.018} \pm 0.001$ | $0.072 \pm 0.005$ | $\mathbf{0.702} \pm 0.015$ | $0.740 \pm 0.014$ |
| Beta | $0.257 \pm 0.03$ | $\mathbf{0.130} \pm 0.005$ | $0.623 \pm 0.035$ | $\mathbf{0.542} \pm 0.017$ |

Table 7: We compare metrics between the GMM-MLE head, and the Fourier-MLE head with 12 frequencies and no regularization, for every dataset in our toy example. We aggregate metrics over 4 different seeds and report the standard deviation.

### C.4 ARE LLMs RANDOM NUMBER GENERATORS?

**Dataset:** We create a training dataset using the prompt template: *"The following is a list of normally distributed random numbers in the interval [-1, 1] with mean µ and std σ:* $x_1, x_2,$ *"* and response template: *"$x_3$"*, where $(\mu, \sigma) \in \{(-0.55, 0.10), (-0.03, 0.24), (0.42, 0.16), (0.55, 0.10)\}$, and each

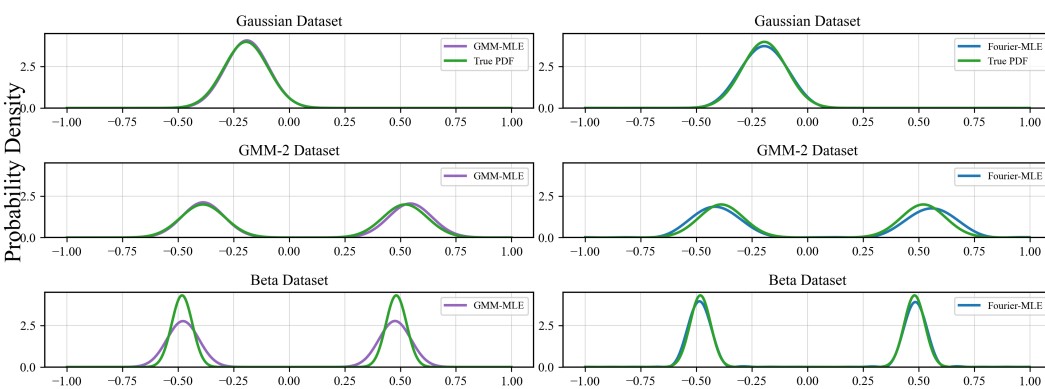

Figure 11: Comparison between the PDFs learned by the GMM-MLE head and the Fourier-MLE head for each of the datasets in the toy example. While GMM-MLE outperforms Fourier-MLE on the Gaussian and GMM-2 datasets, Fourier-MLE performs better on the Beta dataset.

$x_i \sim \mathcal{N}(\mu, \sigma)$. We write each number using two decimal places. Our training dataset consists of 256 such (prompt, response) pairs, divided evenly among the four distributions.

**Model Architecture:** We consider three different models: the original Llama-3.1-8B-Instruct model Dubey et al. (2024), the original model after LoRA fine-tuning, and the original model where we replace the linear classification head with the Fourier head and perform LoRA fine-tuning. For the Fourier head, we use an output dimension of 200 and the original latent space $[-1, 1]$ because of our chosen decimal precision. We conduct LoRA fine-tuning for 16 epochs with a learning rate of $3 * 10^{-4}$ and a linear decay schedule, and a batch size of 64. We release all of our training code on our project page.

**Model Evaluation:** We compute two metrics: the first is Total Variation Distance; we define this to be one half of the $L^\infty$ distance between the ground truth quantized Gaussian histogram, and the empirical histogram of samples, quantized into 20 bins. Our second metric is the Quantity of Unique Samples. In Figure 3 we present example histograms for each of the three models that we consider for the median TVD in each class. Those results demonstrate that the Fourier head learns a more accurate PMF. And in Figure 12 we demonstrate that the Fourier head model consistently obtains a lower TVD and a greater diversity of samples. We hypothesize that the LoRA-finetuned baseline model has fewer diverse samples because it memorizes training data instead of learning the actual distribution. In contrast, the Fourier head is forced to learn a continuous distribution, and samples directly from that distribution.

**Related works which consider LLMs as random number generators:** Gu et al. (2024) explores probability distribution sampling in LLMs in the context of behavioral simulation, which demonstrates an application of using LLMs as random number generators. And Paruchuri et al. (2024) explores the tasks of using LLMs for estimating percentiles, drawing samples, and calculating probabilities. Lastly, Hopkins et al. (2023) explores sampling from numerical probability distributions using LLMs, but they only consider a single non-uniform density, namely the normal distribution $N(0.5, 0.2887)$. They find that LLMs don't perform well at this task, but they don't thoroughly investigate model-based interventions. In contrast, we thoroughly investigate the impact of fine-tuning, and replacing the token classification head one with a better inductive bias for modeling complex probability distributions.

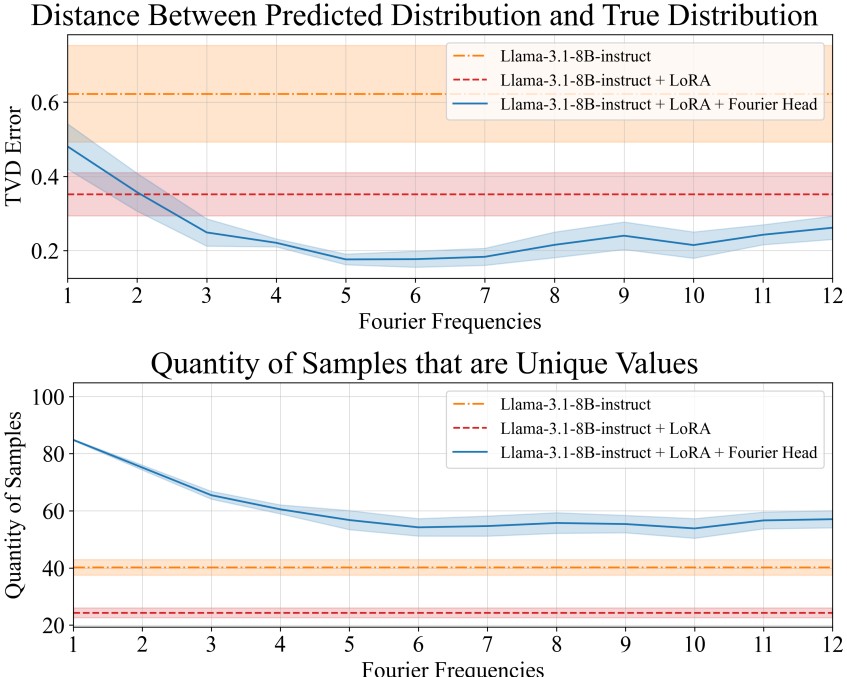

Figure 12: We demonstrate that the Fourier head model consistently obtains a lower total variation distance, as well as a greater diversity of samples. For TVD (top), lower is better, because lower values indicate that the learned distribution is closer to the ground truth distribution. And for quantity of samples (bottom), higher is better, because lower values indicate that the LM has just memorizes specific numbers instead of performing sampling. We present here the mean values across the four distributions, for all ten seeds. We can see that the Fourier head obtains more diverse samples, and learns a distribution closer to the ground truth.

# D    ADDITIONAL EXPERIMENT DETAILS, LARGE-SCALE EXAMPLES

## D.1    DECISION TRANSFORMER

Following the original Decision Transformer implementation, we trained on 500k transitions observed by a DQN agent during training, for 5 epochs. We trained on the same model size as the original implementation (a GPT-1 model with approximately 2.012M parameters) which takes about 4 hours on a single GPU. We can see that in Figure 13 that the PMFs learned by the Fourier head are smoother. In Figure 16 we present results for more Atari games. In Figure 14, we present results from an ablation study of the model size. The results demonstrate that, across model sizes, the Decision Transformer with a Fourier head is better at learning high-quality next action distributions than the Decision Transformer with a linear head. And in Figure 15, we present results from an ablation study of the dataset size, which show that the Fourier head obtains larger returns than the Linear classification head across dataset sizes.

## D.2    CHRONOS

In Figure 17 we present a learned next-token PMF from a linear Chronos model, and a next-token PMF from a Chronos model which uses the linear head. The Fourier head is about 4x smoother. In Table 8 we present results from an ablation study on the quantity of Fourier frequencies, choice of regularization, and binning strategy. We followed the original Chronos implementation, keeping all hyperparameters the same. In particular, we trained for 200k steps, on the same model size as the original implementation (the T5 model with approximately 20M parameters) and this takes about 48 hours on 8 GPUs. See Table 9 for the datasets we used to train and evaluate Chronos.

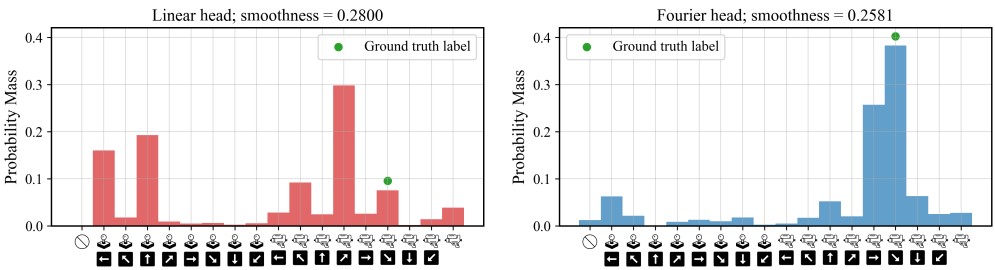

Figure 13: We present example next action distributions for a single step in the Decision Transformer test split. The Fourier agent with 8 frequencies produces a "clump" of actions that is semantically meaningful. Namely, this agent almost certainly wants to shoot in the down right or right direction, presumably because there is a submarine in that direction. In contrast, the linear agent's next-action distribution doesn't clearly depict a strategy, and incorrectly assigns higher likelihoods to incorrect actions. Because the Fourier head outputs a smoother PMF, it learns to concentrate more probability mass near the correct action.

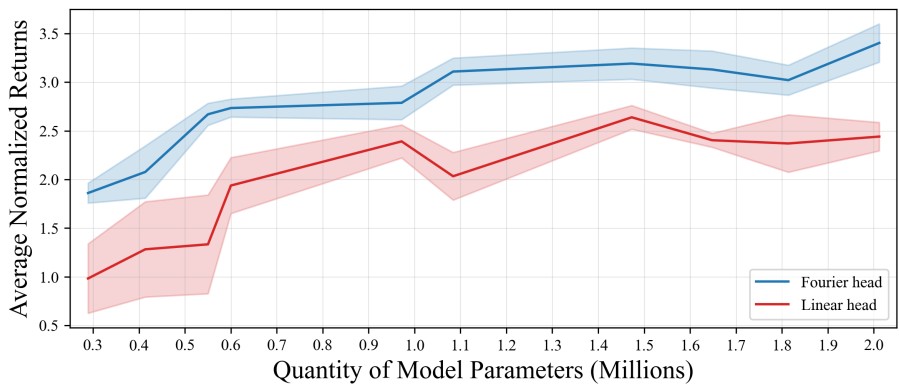

Figure 14: We present an ablation study on the effect of the model size on the relative performance of the Fourier head and the Linear head. The results demonstrate that, across model sizes, the Decision Transformer with a Fourier head is better at learning high-quality next action distributions than the Decision Transformer with a linear head.

Fourier Head Ablation Study: Dataset Size

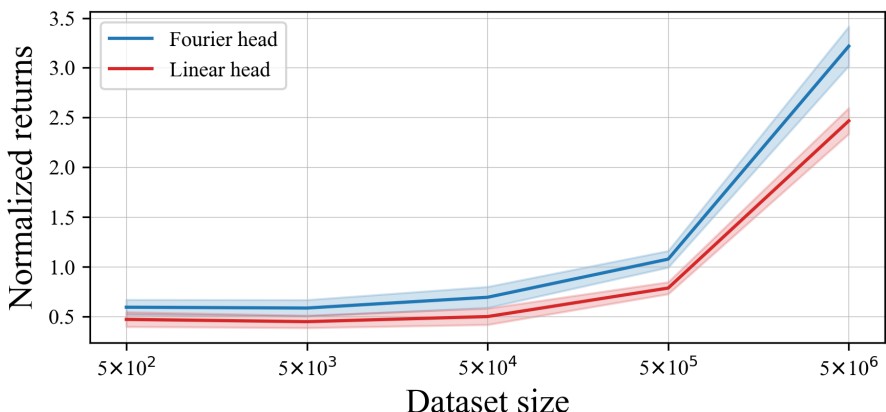

Figure 15: In this ablation study, we analyze whether dataset size has any effect on the relative performance of the Linear head and the Fourier head. Our results show that, across dataset sizes, the Decision Transformer agent with a Fourier head achieves larger returns than the linear head on the Seaquest game.

| Chronos Time Series Model | MASE (↓) | WQL (↓) | Smoothness (↓) |
|---|---|---|---|
| Linear | 0.883 | 0.750 | 0.1689 ± 0.1087 |
| Fourier-64 | 0.875 | 0.798 | **0.0032 ± 0.0012** |
| Fourier-128 | 0.872 | 0.767 | 0.0068 ± 0.0035 |
| Fourier-256 | 0.859 | 0.755 | 0.0139 ± 0.0087 |
| **Fourier-550** | **0.852** | **0.749** | 0.0283 ± 0.0224 |
| Fourier-550 (no regularization) | 0.861 | 0.753 | 0.0286 ± 0.0219 |
| Fourier-550 (uniform precision binning) | 0.873 | 0.747 | 0.0395 ± 0.0252 |

Table 8: We present large-scale experiments on Chronos time series forecasting. Notably, every Fourier model outperforms the linear baseline on MASE and smoothness metrics. We can see that within the Fourier model class, decreasing the number of frequencies lets you trade off the continuity of the learned probability mass functions (smoothness) for the quality of the forecasts (MASE, WQL). In the bottom two rows, we present an ablation for our large-scale experiments on Chronos time series forecasting. The best overall performing Fourier-550 model uses Fourier regularization and mixed precision binning, which are both techniques informed by Fourier analysis. We observe that both of these interventions improve the MASE, but have minimal effect on the WQL. Note that the choice of binning strategy doesn't affect the performance of the linear baseline.

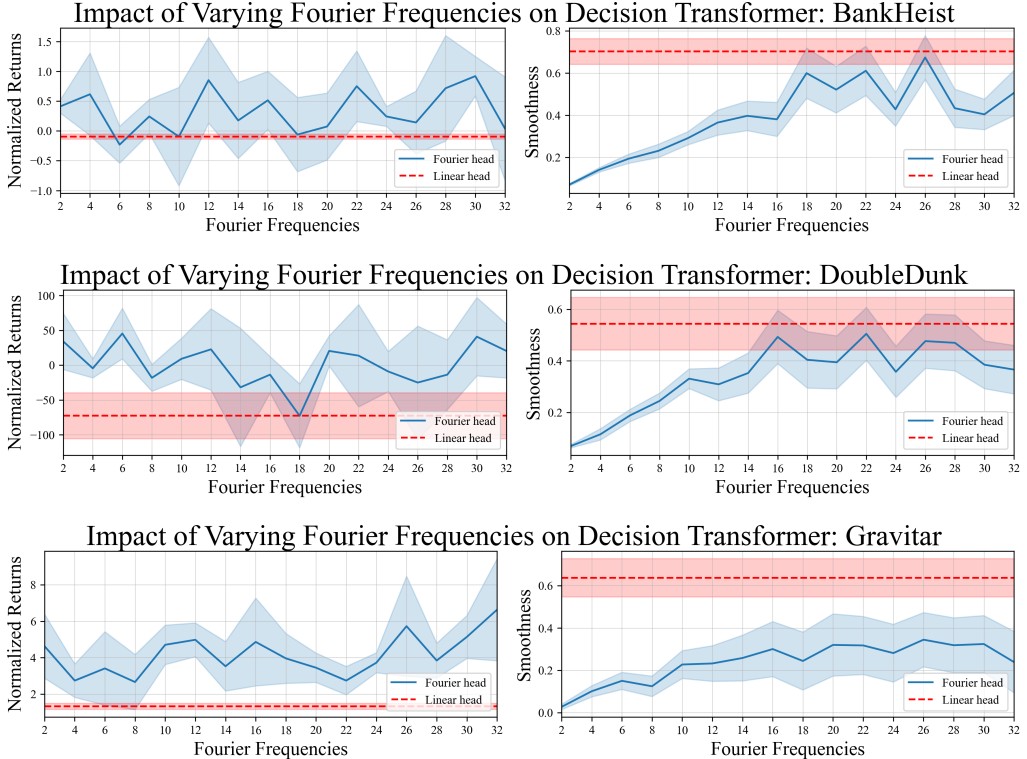

Figure 16: We present empirical results for how the quantity of Fourier frequencies impacts returns and smoothness for additional imitation learning games. For normalized returns, higher is better; for smoothness, lower is better. We can see that for the BankHeist, DoubleDunk, and Gravitar games, the Fourier agent consistently achieves higher normalized returns than the linear baseline agent, while still learning smoother next-action distributions.

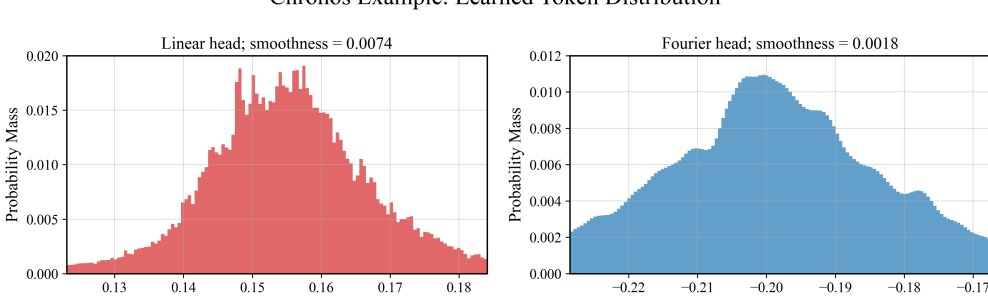

Figure 17: We present the next token value distribution for a single forecasted timestep on the Tourism Monthly dataset. We observe that the Fourier head's learned conditional distribution is smoother, fitting signal more robustly, whereas the linear head overfits to the noise, and is therefore more jagged. We note that the $x$-axis represents the bins in the latent space $[-1, 1]$; the $x$-axis values for the Fourier head are lower because the linear head uses uniform binning, and the Fourier head uses mixed precision binning.

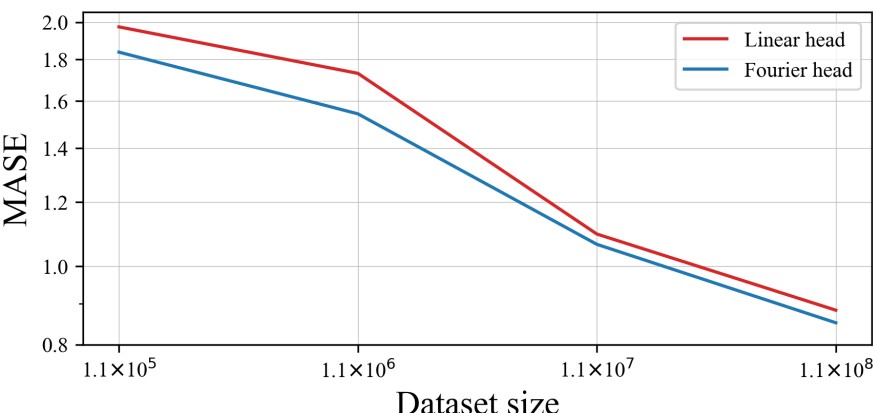

Figure 18: In this ablation study, we analyze whether dataset size has any effect on the relative performance of the linear head and the Fourier head for the probabilistic time series task. Our results show that, across dataset sizes, the Fourier head yields more accurate forecasts than the linear head. For the dataset sizes $1.1 \times 10^5, 1.1 \times 10^6$, and $1.1 \times 10^7$, we report the average MASE across four seeds; for the dataset size $1.1 \times 10^8$ we report the MASE from Table 3. We generate the plot following (Kaplan et al., 2020) and observe a similar power-law scaling behavior for both methods, with the Fourier head consistently outperforming the linear head.

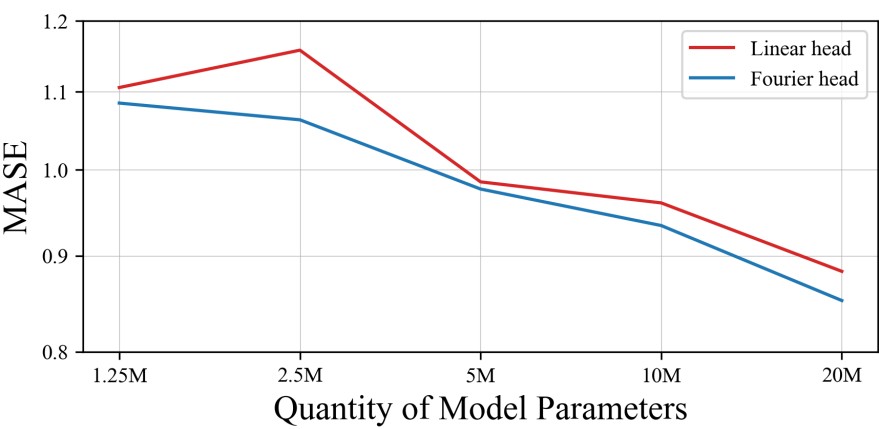

Figure 19: In this ablation study, we analyze whether model size has any effect on the relative performance of the linear head and the Fourier head for the probabilistic time series forecasting task. Our results show that, across model sizes, the Fourier head yields more accurate forecasts than the linear head. For the model sizes 1.25M, 2.5M, 5M, and 10M, we report the average MASE across three seeds; for the model size 20M we report the MASE from Table 3. We generate the plot following (Kaplan et al., 2020) and observe a similar power-law scaling behavior for both methods, with the Fourier head consistently outperforming the linear head.

Table 9: All datasets that are used for our time series forecasting experiments. We built our time series forecasting experiments on top of Chronos (Ansari et al., 2024), and this table is mostly copied from their paper. The datasets are partitioned according to how they are used for training and evaluation of models: *pretraining-only* data is only used for training; *evaluation* data is *not* used in training models, but only for evaluation (final $H$ observations). All of our evaluation datasets came from the zero-shot evaluation set from Chronos.

| Dataset | Domain | Freq. | # Series | Series Length | | | Prediction Length ($H$) |
|---|---|---|---|---|---|---|---|
| | | | | min | avg | max | |
| **Pretraining** | | | | | | | |
| Brazilian Cities Temperature | nature | M | 12 | 492 | 757 | 1320 | - |
| Mexico City Bikes | transport | 1H | 494 | 780 | 78313 | 104449 | - |
| Solar (5 Min.) | energy | 5min | 5166 | 105120 | 105120 | 105120 | - |
| Solar (Hourly) | energy | 1H | 5166 | 8760 | 8760 | 8760 | - |
| Spanish Energy and Weather | energy | 1H | 66 | 35064 | 35064 | 35064 | - |
| Taxi (Hourly) | transport | 1H | 2428 | 734 | 739 | 744 | - |
| USHCN | nature | 1D | 6090 | 5906 | 38653 | 59283 | - |
| Weatherbench (Daily) | nature | 1D | 225280 | 14609 | 14609 | 14610 | - |
| Weatherbench (Hourly) | nature | 1H | 225280 | 350633 | 350639 | 350640 | - |
| Weatherbench (Weekly) | nature | 1W | 225280 | 2087 | 2087 | 2087 | - |
| Wiki Daily (100k) | web | 1D | 100000 | 2741 | 2741 | 2741 | - |
| Wind Farms (Daily) | energy | 1D | 337 | 71 | 354 | 366 | - |
| Wind Farms (Hourly) | energy | 1H | 337 | 1715 | 8514 | 8784 | - |
| **Evaluation** | | | | | | | |
| Australian Electricity | energy | 30min | 5 | 230736 | 231052 | 232272 | 48 |
| CIF 2016 | banking | 1M | 72 | 28 | 98 | 120 | 12 |
| Car Parts | retail | 1M | 2674 | 51 | 51 | 51 | 12 |
| Hospital | healthcare | 1M | 767 | 84 | 84 | 84 | 12 |
| M1 (Monthly) | various | 1M | 617 | 48 | 90 | 150 | 18 |
| M1 (Quarterly) | various | 3M | 203 | 18 | 48 | 114 | 8 |
| M1 (Yearly) | various | 1Y | 181 | 15 | 24 | 58 | 6 |
| M3 (Monthly) | various | 1M | 1428 | 66 | 117 | 144 | 18 |
| M3 (Quarterly) | various | 3M | 756 | 24 | 48 | 72 | 8 |
| M3 (Yearly) | various | 1Y | 645 | 20 | 28 | 47 | 6 |
| M4 (Quarterly) | various | 3M | 24000 | 24 | 100 | 874 | 8 |
| M4 (Yearly) | various | 1Y | 23000 | 19 | 37 | 841 | 6 |
| M5 | retail | 1D | 30490 | 124 | 1562 | 1969 | 28 |
| NN5 (Daily) | finance | 1D | 111 | 791 | 791 | 791 | 56 |
| NN5 (Weekly) | finance | 1W | 111 | 113 | 113 | 113 | 8 |
| Tourism (Monthly) | various | 1M | 366 | 91 | 298 | 333 | 24 |
| Tourism (Quarterly) | various | 1Q | 427 | 30 | 99 | 130 | 8 |
| Tourism (Yearly) | various | 1Y | 518 | 11 | 24 | 47 | 4 |
| Traffic | transport | 1H | 862 | 17544 | 17544 | 17544 | 24 |
| Weather | nature | 1D | 3010 | 1332 | 14296 | 65981 | 30 |

