# OpenReview forum: "Fourier Head: Helping Large Language Models Learn Complex Probability Distributions"
_ICLR.cc/2025/Conference — ICLR 2025 Poster_

### Official Review · Reviewer_WB6H · 2024-10-20

**Soundness:** 3
**Presentation:** 3
**Contribution:** 3
**Rating:** 8
**Confidence:** 3

**Summary:**

The paper proposes the "Fourier head" as an alternative to linear classification heads for tasks where the output can be thought of as a quantization of a continuous space. The Fourier head learns the Fourier coefficients of the target function and then quantizes it instead of learning the quantized values directly. The authors theoretically and empirically demonstrate that varying the number of frequencies in the Fourier head trades off between smoothness and modelling performance. The Fourier head is shown to improve over the baseline linear head on a toy task, an agentic decision-making task, and for time-series modelling.

**Strengths:**

- The paper provides evidence that the Fourier head is an improvement over the baseline linear head in a wide variety of settings (toy example, agentic decision making, time-series modeling).
	- The Fourier head improves both smoothness and accuracy (MLE, MASE, WQL).
- The exposition of the Fourier head is clear and easy to understand.
- Various practical details (training objective, hyperparameter choice, regularization, binning strategy) are provided. This is helpful for reproducibility and to those who wish to apply the Fourier head to other tasks.

**Weaknesses:**

- For regression tasks with continuous-valued target output, it is not clear to me the practical motivation for outputting an entire probability distribution, instead of just a point estimate. Thus one of the main advantages of the Fourier head, that its outputs are more smooth, feels somewhat unmotivated to me. I would like to see more discussion of why exactly the smoothness is beneficial in practice.

**Questions:**

- The justification of the Fourier regularization term as imposing that the coefficients decay as $1/n^2$ is a little strange to me -- this is an asymptotic condition and, in practice, there are a finite number of coefficients, so isn't the condition always vacuously met?
- For the Decision Transformer, the output space as described in the paper is more naturally a quantization of $S^1 \sqcup S^1\sqcup \\{0,1\\}$ instead of $[-1, 1]$. (Either a shooting direction or a moving direction, each of which takes eight different values arranged on the circle $S^1$. Also two actions without an associated direction.) It would be interesting to see if the Fourier head can be generalized to output spaces that are not naturally interval-shaped.
- Actually, if I remember correctly, functions can only be approximated by Fourier series if they are periodic, i.e. functions on $S^1$. I suppose this does not affect the toy example and the time-series modelling, since the interval is chosen to be large enough that the distribution is near zero at the boundaries and so is approximately periodic. But I wonder if this is a limitation in other settings.
- Often, for tasks with continuous-valued target output (e.g. the toy example and time-series example), only a point estimate is necessary, not the full distribution. Hence a good baseline to include for the toy example is an MLP model with only one output dimension (possibly with atan nonlinearity to map outputs to the interval), evaluated on MSE. Likewise for the time-series example, but with MASE.

Minor typos:
- Line 193: "hyperparamter" -> "hyperparameter"
- Line 205: $c_n$ should be $c_k$
- Line 244: "and $D$ be some measure of discrepancy such that $L^2$," should be "such as"?
- Line 257: "Denote by $g_\sigma(x)$ is" delete "is"
- Line 515: "descretized" -> "discretized"

---

> ### Author Response · Authors · 2024-11-21
> **Author response**
>
> Thank you for the thoughtful feedback! We were able to produce the additional results that you asked for, summarized here:
>
> * We introduce a new pointwise regression baseline model, and demonstrate that the Fourier classification head outperforms it.
> * We provide more evidence that the Fourier head can model latent spaces which are not circle-shaped, by adding results for three more RL tasks where the latent space is a quantization of $S^1\sqcup S^1\sqcup \\{0,1\\}$. In all these tasks, the Fourier head agent obtains significantly larger returns than the baseline.
>
> Below, we provide more details, as well as question-specific responses. We have also uploaded a revised manuscript, with changes written in blue.

---

> ### Author Response · Authors · 2024-11-21
> **Author response**
>
> ## Q1: for tasks with a continuous-valued target output, does it make more sense to output a point estimate?
>
> **We have conducted additional experiments, and our results show that the probabilistic nature of the tasks leads to a pointwise regression model “regressing to the mean” and performing poorly on the MSE metric.**
>
> In more detail: we have created another toy example baseline model with the same architecture as the Fourier head model, but where we substituted the last classification layer with a linear layer having a single output dimension, and where we train using MSE loss. We refer to this model as “Pointwise Regression”. When we evaluate each of Pointwise Regression, Fourier head, and Linear head on the toy experiment datasets, their average MSE performances across seeds are all approximately equal to each other, showing no observable advantage in terms of MSE for one model over another.
> This is to be expected, given the probabilistic nature of the datasets. We include here our new results:
>
> |          | Pointwise Regression | Linear Classification | Fourier Classification |
> |----------|-------------------|---------|----------|
> | Gaussian | 0.010 ± 0.001 | 0.011 ± 0.001 | 0.012 ± 0.001 |
> | GMM-2    | 0.121 ± 0.004 | 0.126 ± 0.004 | 0.123 ± 0.005 |
> | Beta     | 0.275 ± 0.009 | 0.276 ± 0.008 | 0.275 ± 0.008 |
>
> To demonstrate why a simple pointwise estimate in a probabilistic setting might be inadequate, we also created a new dataset (the "Beta" dataset in the above table) for which the conditional distributions are symmetric about $0$ by construction. More precisely, the conditional distribution of $z$ given $(x,y)$ is $b * \mathrm{Beta}(100 |x|, 100|y|)$, where $b$ is a random variable that is $+1$ or $-1$ with equal probability. We have used the Beta distribution since it is a non-Gaussian distribution, so it illustrates the flexibility of the Fourier head in learning a wide variety of naturally occurring complicated distributions. **In fact, on the Beta dataset, our results show that the Pointwise Regression, Linear classification head, and Fourier head models all have identical MSE very close to 0.275, which is also the MSE of a naive baseline model that always predicts 0 regardless of the input.** In other words, the Pointwise Regression model collapses to always predicting values very close to 0 since it regresses to the mean of the underlying distribution, completely missing any information about higher moments of the distribution.
>
> On the other hand, the Fourier head is able to learn a high quality reconstruction of the underlying conditional distribution, showing why one might not choose to rely on a pointwise estimate even in our toy setting. We have included the Pointwise Regression baseline MSE values in the latest draft.
> We also include visual samples of the model predictions alongside the true conditional distribution, illustrating this unwanted "regression to the mean" behavior that arises. [(link to graph)](https://drive.google.com/file/d/1ikH_DDJcVCTGslCzKxvSlYL326g8tpK2/view?usp=sharing)
>
> Given these results in the toy setting, we believe that using a classification head that can generate a probabilistic forecast rather than a pointwise regression in the Chronos experiments is desirable since the underlying next-token distributions are complicated and not well-captured by regressing to the mean.
>
> (On a technical note: your suggestion prompted us to realize that for evaluating the MSE performance of the categorical distributions predicted by the Fourier and Linear head, one should consider their expected values as the model’s implied pointwise estimate, rather than just the bin of maximum probability. This is especially applicable since some of our toy distributions are bimodal. As a result, we have updated our MSE metrics in the paper. So, thank you for asking this question.)

---

> ### Author Response · Authors · 2024-11-21
> **Author response**
>
> ## Q2: why should we impose the $1/n^2$ Fourier regularization term during training, if it’s a truncated Fourier series?
>
> If the amplitudes of a truncated Fourier series decay like $1/n^2$, this ensures that the function is smoother.
>
> In more detail–in the presence of limited data, there are many possible choices of Fourier coefficients which may fit the data equally well. We can express a preference towards smoother densities by penalizing the higher-order Fourier coefficients more than the lower-order Fourier coefficients. **Intuitively, this ensures that the model extracts low-frequency signal from the training data, while ignoring high-frequency noise.** We’ve added a summary of this in Section 2.4 of the revision. Thanks for this question, it seems like we didn’t make this point clear enough in the submission.

---

> ### Author Response · Authors · 2024-11-21
> **Author response**
>
> ## Q3: Can the Fourier head be generalized to output spaces which are not interval shaped?
>
> This is certainly possible, but beyond the scope of the paper. One way to do this would be to have the learned output categorical distribution be a mixture of Fourier heads and a linear classification head. For example, since the action space for our RL task is a quantized version of $S^1 \sqcup S^1 \sqcup \\{0, 1\\}$, this hypothetical architecture would learn two Fourier heads, each with output dimension 8, as well as a linear classification head, with output dimension 2. Such a model would also have a 3-dimensional classification layer which functions as a high level controller, choosing which of the three classification heads to route the model input to.  It would need to be trained with masking to ensure that gradients only update for the correct head.
>
> We don’t conduct experiments on this generalized classification head for two reasons: first, it is sufficiently complicated that it is beyond scope, and we hope that future works will be inspired to make this applicable to those other domains. And second: our Decision Transformer results show that a single Fourier classification head does a much better job than the linear baseline, even while incorrectly placing a continuity inductive bias between some of the token boundaries, as your review noted. We believe that the positive results, despite this limitation, underscores just how problematic it is that many models continue to use a linear classification head, while the next-token distribution seems to be “starving” for at least some continuity inductive bias. To demonstrate this more robustly, we have added results for three more Atari games. **In addition to our previously reported results on Seaquest, the paper now includes results on BankHeist, DoubleDunk, and Gravitar, which demonstrate that the Fourier head significantly outperforms the baseline.**
>
> | Classification Head | BankHeist | DoubleDunk | Gravitar | Seaquest |
> |-------------------|------------|-------------|-----------|-----------|
> | Linear head |-0.09 $\pm$ 0.05 | -72.72 $\pm$ 33.08 | 1.32 $\pm$ 0.17 | 2.53 $\pm$ 0.63 |
> | Fourier head | **0.92 $\pm$ 0.33** | **45.45 $\pm$ 36.36** | **4.98 $\pm$ 0.93** | **3.70 $\pm$ 0.47** |
>
> In this table, we report normalized returns (mean $\pm$ standard deviation, averaged over four seeds) for the Decision Transformer agent across the four Atari games. The results show that the Fourier agent obtains higher returns than the Linear agent across all games.

---

> ### Author Response · Authors · 2024-11-21
> **Author response**
>
> ## Q4: Is it a limitation that the Fourier series can only model periodic functions?
>
> In our large-scale experiments we find that this isn’t an issue, for the very reason that you mentioned–namely, for the time series task the distribution is close to zero near the boundaries. **If needed, a tanh reparameterization can be used to map the domain $[-1,1]$ to the real line, and you can then truncate it. This would let the Fourier head learn non-periodic functions.** We have added an explanation about this in Section 2.4 of the updated manuscript.
>
> However, maintaining the original periodic formulation and using a Fourier head with a sufficiently large number of frequencies tends to solve the problem. This follows from the Fourier theory: if you have a smooth function on $[-1,1]$, but $f(-1)\neq f(1)$, then you can learn a Fourier series that does a very good job approximating $f$ over the open interval $(-1,1)$, and it will only have problems near the boundaries $\pm 1$. Using more frequencies guarantees that the approximation improves close to the boundaries.

---

> ### Author Response · Authors · 2024-11-21
> **Author response**
>
> ## Q5: the minor typos
>
> Thanks for catching those, we’ve fixed them in the latest draft :)

---

> ### Author Response · Authors · 2024-11-25
> **Author follow-up**
>
> We wanted to follow up to make sure that your concerns are being properly addressed. Please let us know if there are additional questions that we can provide further information / experiments on. Thank you again for your time and effort reviewing our paper!

---

> ### Comment · Reviewer_WB6H · 2024-11-26
> **Additional comments**
>
> Thank you for your detailed response and additional experiments. I have some remaining questions/comments:
>
> ### $1/n^2$ regularization
> The explanation of the regularization is still not entirely clear to me. You write:
> > for the class of Fourier series which have continuous second derivatives, the Fourier coefficients decay on the order of $1/n^2$. To impose this regularity assumption on the learned Fourier densities, we [...] add a regularization term to the loss
>
> On first reading, I understood this to mean that the $k^2$ factor in the regularization $\sum_k k^2 |c_k|^2$ is used to enforce that the coefficients decay like $1/n^2$ (and this causes the Fourier head output to have continuous second derivatives?). However, it's unclear to me what this could mean in the case of truncated Fourier series. For instance, the condition that $c_n\in O(1/n^2)$ is an asymptotic statement; when the Fourier series is truncated, it vacuously holds regardless of whether the regularization is present. If you had some non-asymptotic interpretation of $c_n\sim 1/n^2$ in mind, I don't know what it could be or why the $k^2$ regularization would cause it to hold.
>
> I find the motivation in de la Fuente (2024) II.D to be much more clear -- the regularization term is equal to the total squared variation (Eq 10), which is a measure of smoothness.
>
> Also: it seems that smoothness could be increased either by increasing regularization strength or decreasing the number of Fourier frequencies $N$. How do these relate? When is it appropriate to tune one vs the other?
>
> ### Exposition of the Fourier head and the de la Fuente (2024) paper
> Additionally, in my opinion [de la Fuente et al (2024)](https://arxiv.org/abs/2402.15345) does a better job of explaining some other details:
> - The purpose of first learning autocorrelation coefficients then converting to Fourier coefficients (Alg. 1 Step 4) is explained in their section II.A
> - The normalization by $1/c_0$ (Alg. 1 Step 5) is explained in their II.B.
>
> It would be helpful to briefly mention these to help explain Algorithm 1. (Or maybe just refer the reader to the sections in their paper.)
>
> Overall, it appears that your work takes significant inspiration from the de la Fuente paper. Your paper cites theirs at several places, which is good, but it might also be appropriate to mention them in the intro and/or when the Fourier head is introduced in Sec 2.1/2.2. My understanding is that the Fourier head is effectively the "Fourier basis density model" that they propose swapped in for the last layer of a deep neural model.
>
> Apologies for not bringing this up earlier -- I hadn't looked at the de la Fuente paper at the time.
>
> ### Point estimate vs probabilistic forecasting
> Thank you for the pointwise regression experiments. Intuitively, the fact that the pointwise regression predicts the mean even for bimodal distributions is not ideal. However, if one only cares about MSE, then your experiments show that the point estimate is perfectly fine. The paper would be better motivated if you could mention some examples of when the MSE alone is insufficient, and predicting the full density is superior to just the point estimate. It looks like the RL setting is one such example, since the agent needs to explicitly sample from the output distribution. (Maybe this should be made more explicit in the paper.) Are there similar examples for the time-series setting?

---

> ### Author Response · Authors · 2024-11-26
> **Author response**
>
> We appreciate the reviewer’s engagement with us! In response to each point:
>
> ---
>
> ### $1/n^2$ regularization
>
> We agree that interpreting the regularization term as the total squared variation is much more natural given its relation to smoothness. We have modified the manuscript to reflect this change (see e.g. updated Section 2.4).
>
> As for choosing whether to increase regularization strength or decrease the number of Fourier frequencies: tuning the regularization strength gives finer control on penalizing the model for high frequency content to increase smoothness, while still allowing some high frequency information. The number of frequencies is a more drastic handle on this tradeoff, as made explicit in our scaling law. Choosing which to tune needs to be determined for each application – for applications in which the underlying distributions are representable in a few number of frequencies $N_0 \ll m/2$, minimizing the number of frequencies until $N_0$ might be preferable in order to increase smoothness. On the other hand, in applications where a high number of frequencies are required to obtain a reasonable reconstruction of the distribution, we would prefer to maximize the number of frequencies while tuning the regularization strength in order to encourage smoothness while not sacrificing modeling capacity.
>
> ---
>
> ### Exposition of the Fourier head and the de la Fuente (2024) paper
>
> As you noticed, we only tried to provide a minimal exposition of the Fourier Basis Density Model paper. Going over it now, we agree with you that our exposition was too minimal, so we’ve added additional references and explanations--
>
> * In the introduction, we’ve added the sentence: *The Fourier head is constructed using the Fourier Basis Density Model (de la Fuente et al, 2024).*
> * When we define Fourier head algorithm, we’ve added the sentence: *The Fourier head is constructed using the Fourier Basis Density Model from (De la Fuente et al., 2024). For more details on the original method (e.g. justification for how learning the autocorrelation coefficients guarantees that the Fourier series has integral 1, and justification for normalizing the Fourier coefficients by $\mathrm{Re}(c_0)$), we refer the author to (De la Fuente et al., 2024).*
>
> ---
>
> ### Point estimate vs probabilistic forecasting
>
> In the paper, we only consider scenarios where probabilistic modeling is necessary, and where a point estimate is insufficient. We constructed the toy example with this in mind–the main success metric for the toy example is the KL divergence between the quantized ground truth distribution, and the learned categorical distribution. Similarly, the large-scale examples in the paper (probabilistic agentic decision making, and probabilistic time series forecasting) require learning a probability distribution over the latent space, and sampling from it at test time to obtain the success metrics.
>
> In particular, **probabilistic time series forecasting is a useful tool for decision making because probabilistic forecasts allow us to precisely quantify future uncertainty.** We note that there are tradeoffs when deciding whether to model time series probabilistically versus deterministically:
>
> * *Deterministically* modeling time series (e.g. learning to regress, with an MSE loss) is generally simpler, especially during data preprocessing. For example, tokenization is not needed.
>
> * *Probabilistically* modeling time series (e.g. learning a distribution over the next value of the time series, using cross entropy loss, as in Chronos) is more complicated, as it requires design choices such as tokenization. But the upshot of these methods is that probabilistic forecasts contain all the information from deterministic forecasts, plus more. For example, from a probabilistic time series model, you can choose to sample many possible futures. Computing the median of those futures allows you to compute an accuracy metric like MSE. Additionally, you can extract error bars using the quantiles from the possible futures, which is a clear advantage for practical applications.
>
> And as you requested, we have added explicit descriptions for our large-scale tasks to make it clear that they involve probabilistic sampling--
> * For the RL task, we added: *“ At test time, the agent chooses its next action by sampling from the learned next-action distribution.”*
> * For the time series task, we added: *“At test time, the model chooses the next numerical token by sampling from the next-token distribution.”*
>
> Please let us know if you have any further questions, or if there is anything that we can clarify!

---

> > ### Comment · Reviewer_WB6H · 2024-11-26
> >
> > Thank you for making the requested writing changes. I've updated my score.

---

> ### Author Response · Authors · 2024-12-03
> **Following up**
>
> We just wanted to follow up, and say thank you for your effort reviewing our paper, and for your engagement throughout the rebuttal period!

---

### Official Review · Reviewer_aPiu · 2024-10-31

**Soundness:** 3
**Presentation:** 3
**Contribution:** 3
**Rating:** 6
**Confidence:** 4

**Summary:**

The authors argue that current methods for parameterizing a discrete distribution over numerical data suffer from ignoring ordinal structure, which should imply that adjacent discrete buckets will have similar density and therefore "smoothness" in the probability mass function. To fix this oversight, the authors propose a new parameterization on the coefficients of a Fourier series, leading to a smooth function on the interval [-1,1], which is then quantized. The new parameterization is therefore a drop-in alternative to a uniform bucketing of the interval. The method is evaluated on toy univariate densities as well as on an offline reinforcement learning problem and in time-series forecasting, and the result indicate that using Fourier head leads to lower errors in density estimation and higher returns in reinforcement learning.

**Strengths:**

The proposed method is simple and outlined with clarity in the paper. While the method is not complex, it is relatively novel to my knowledge. The significance is also reasonably large because modeling continuous numerical values using discrete tokens is increasingly popular.

**Weaknesses:**

To my understanding, the main goal of the paper is to propose a practical method, and given this goal, the empirical evaluation is not very impressive. I'll break this criticism down into a few subcategories

1. Emphasis on smoothness: The authors devote a lot of space and attention to the notion of "smoothness", proposing a new metric to measure it and including this metric in all the evaluations. However, from a practical standpoint it's not clear why we should care about smoothness independent of its effect on metrics like MAE or RMSE. In fact, it's possible to contrive examples where we want less smoothness (related to the square wave examples in the appendix), and it's not clear a priori that the marginal distributions for a particular downstream application will be "smooth". The "smoothness" numbers therefore feel like a distraction from what really matters, which is whether this ordinal inductive bias actually helps the model fit the data distribution. In many cases, the method seems to improve smoothness without affecting reward/loss or vice versa.

2. Limited empirical impact: while Fourier head does seem to yield significant benefits in offline RL, it doesn't seem to have a significant effect on time series modeling. The benefit in terms of MASE and especially in terms of WQL is very marginal, and if I were looking for ways to improve my model, I might not adopt the additional complexity needed for such a small improvement, which is probably on par with tuning other hyperparameters or making small architectural changes. It might be helpful to identify possible explanations for why the effect is relatively minor in time series but more pronounced in offline RL. For example, are the next-action distributions significantly different in their multimodality? It might be much more compelling to replace the time series experiments with additional offline RL experiments if that application happens to be the ideal use case for this method.

3. Limited baselines: Fourier head is only compared to the most naive possible baseline, uniform binning on [-1, 1]. In practice, there are more widely-used alternatives, such as Gaussian mixture models (GMMs) and quantile regression. Both of these techniques have an ordinal bias and should learn solutions that are much more smooth. I don't know if these methods are viewed as out-of-scope in this paper because they are not learned with cross-entropy loss. From one perspective, it might be reasonable to limit the investigation to discrete tokenization methods and discrete loss functions, but it does make the practical impact lower, as it's hard to tell whether this method is actually the best among all simple options or just an improvement upon simple uniform binning. This particular subcategory of criticism feels especially pertinent given the toy experiments in Section 4, where Fourier head is shown to approximate a GMM. It seems reasonable to conclude that in many cases a GMM should also therefore be able to approximate Fourier head. Is the converse not true and how important are the case where GMMs might not be able to match the performance of Fourier head?

Beyond empirical evaluation I think there are also other potential weaknesses:

1. Limited expressiveness: Presumably this method only works for bounded sequences. In the case of RL this might be reasonable if the state and action spaces are constrained. In the case of time series, this limits applications to series without a significant trend component, which would eventually cause the values to exit the range of past observations.

2. Additional hyper-parameters in the form of chosen Fourier series frequencies and regularization strength.

**Questions:**

I included a few questions in my "Weaknesses" response. I've also included a few below:

1. How were the frequencies used in the time series experiments chosen? Were they chosen a priori or through a cross-validation procedure? If cross-validation, how were the splits constructed?

2. Why not explore other bases besides the Fourier basis? Is there something intrinsically better about that basis? Alternatively, there are many other parameterizations that would encourage smoothness. For example, one could parameterize only the differences between buckets and regularize these differences to be small. The final probability mass function would be calculated by integrating the differences. Is there a reason to believe a priori that this approach might perform worse?

---

> ### Author Response · Authors · 2024-11-21
> **Author response**
>
> Thank you for the thoughtful feedback! We were able to produce the additional results that you asked for, summarized here:
>
> * We demonstrate additional empirical impact by adding results for three more RL tasks. In all these tasks, the Fourier head agent obtains much larger returns than the baseline.
> * We demonstrate additional empirical impact by including two more ablations: one which demonstrates that the benefit of the Fourier head persists as the *dataset size* scales, and one which demonstrates that the benefit of the Fourier head persists as the *model size* scales.
> * We introduce a new GMM baseline model, as well as a more complicated synthetic dataset, and demonstrate that the Fourier head’s added flexibility is needed to model this more challenging dataset.
>
> Below, we provide more details, as well as question-specific responses. We have also uploaded a revised manuscript, with changes written in blue.

---

> ### Author Response · Authors · 2024-11-21
> **Author response**
>
> ## Q1: why all the emphasis on “smoothness”?
>
> A primary goal of the paper was to improve performance on the original metrics from the tasks we considered (e.g. accuracy for time series, returns for RL), and we accomplished that using the Fourier head, while also showing that the learned categorical distributions are smoother for the Fourier head than the linear head.
>
> And you’re absolutely right that, in the context of this paper, smoothness is not a metric that we should care about on its own. Our experiments show that the Fourier head yields smoother categorical distributions than the Linear classification head. But we also find that smoothness of the categorical distribution doesn’t necessarily lead to better downstream performance.
>
>  **To provide additional evidence that the Fourier head provides concrete improvements for the original success metrics, we have added two more ablation studies to the paper:**
>
> * Ablation study #1: We show the Fourier head is better at learning high-quality next action distributions than the Decision Transformer with a Linear head *across all dataset sizes*.
> [(link to graph)](https://drive.google.com/file/d/16qJsSBJ9xwT6PiqYqYco6pBkUiHaXX9x/view?usp=sharing)
> * Ablation study #2: We show the Fourier head is better at learning high-quality next action distributions than the Decision Transformer with a Linear head *across all model sizes*.
> [(link to graph)](https://drive.google.com/file/d/1-mnyWD4F1Rqgj3-xSVenRp17UWGAMGU1/view?usp=drive_link)
>
> In these ablations, we show that when the Fourier head learns a very smooth density, the generalization is better. Intuitively, this is because if the model needs to learn the likelihood of the $N$’th token, it can lean on the learned likelihood of its neighboring $N-1$ and $N+1$ tokens, even if it saw very few examples during training where the $N$’th token was the correct answer.

---

> ### Author Response · Authors · 2024-11-21
> **Author response**
>
> ## Q2: how to demonstrate more empirical impact?
>
> To strengthen our empirical contribution, we have added results for three more Atari games. **In addition to our previously reported results on Seaquest, the paper now includes results on BankHeist, DoubleDunk, and Gravitar, which demonstrate that the Fourier head significantly outperforms the baseline.**
>
> In this table, we report normalized returns (mean $\pm$ standard deviation, averaged over four seeds) for the Decision Transformer agent across the four Atari games. The results show that the Fourier agent obtains higher returns than the Linear agent across all games.
>
> | Classification Head | BankHeist | DoubleDunk | Gravitar | Seaquest |
> |-------------------|------------|-------------|-----------|-----------|
> | Linear head |-0.09 $\pm$ 0.05 | -72.72 $\pm$ 33.08 | 1.32 $\pm$ 0.17 | 2.53 $\pm$ 0.63 |
> | Fourier head | **0.92 $\pm$ 0.33** | **45.45 $\pm$ 36.36** | **4.98 $\pm$ 0.93** | **3.70 $\pm$ 0.47** |
>
> Furthermore, **we also demonstrate that the Fourier head consistently outperforms the Linear head, irrespective of the quantity of Fourier frequencies, for all these additional games.** [(link to graph)](https://drive.google.com/file/d/1TeWuaxUyFN76oaqGDzYZ9wn4Ggq3Ohj1/view?usp=drive_link)
>
> Lastly, while we agree that the paper’s RL results are relatively stronger than the time series forecasting results, we want to underscore that a 3.5% accuracy improvement on a recent SOTA forecasting model is no easy feat. For comparison, in the original Chronos paper, the authors find that their novel architecture beats the next best task-specific model by only 0.9%. Our paper’s 3.5% forecasting improvement didn’t require any hyperparameter changes to the original configuration, making the Fourier head an easy “drop-in” replacement for a performance increase.

---

> ### Author Response · Authors · 2024-11-21
> **Author response**
>
> ## Q3: the toy experiment has limited baselines, and in particular would benefit from a comparison with a Gaussian Mixture Model
>
> To address this concern we have implemented a GMM classification layer for which the means and standard deviations are learned, and the number of Gaussian components is a hyperparameter. As you predicted, our results show that substituting the GMM head in our toy experiment yields better performance than the Fourier head on the previous datasets. This is to be expected since the conditional distributions being learned in those datasets are precisely GMMs.
>
> To highlight the flexibility of the Fourier head over the GMM head, we have added a more challenging dataset to the toy example, where the conditional distributions are based on a Beta distribution.
> **On this new Beta dataset, the Fourier head achieves KL divergence 0.191 while the GMM head (with 2 Gaussians) achieves KL divergence 0.407, more than twice that of Fourier. Since the Beta distribution is a common and naturally occurring distribution, this comparison shows the advantage of using the Fourier head when the underlying next-token distribution might be complicated and unknown.**
>
> We include results with the GMM head and Beta dataset in our latest draft. In particular, our numerical results show that the Fourier head learns the Beta distribution more robustly than both the GMM head and the linear head:
>
> | Dataset   | Linear head (KL)       | GMM head (KL)           | Fourier head (KL)       |
> |-----------|-------------------|--------------------|--------------------|
> | Beta      | 0.234 $\pm$ 0.032     | 0.407 $\pm$ 0.012      | **0.191 $\pm$ 0.016**  |
>
> We also analyze the effect of Fourier frequencies on learning this Beta dataset, and we find that the Fourier head indeed outperforms the other classification heads for sufficiently many frequencies. [(link to graph)](https://drive.google.com/file/d/1Ux_8fOkN98go58AeDJ1SzZweMWyBepwx/view?usp=drive_link)
>
> Lastly, thank you very much for asking this question. Our choice of synthetic datasets in the toy experiment certainly makes it seem that the Fourier head could easily be replaced by a learned GMM.

---

> ### Author Response · Authors · 2024-11-21
> **Author response**
>
> ## Q4: does the Fourier head have limited expressiveness because it is bounded?
>
>
> In practice, we find that this is not an issue. **If needed, a tanh reparameterization can be used to map the domain $[-1,1]$ to the real line. This would let the Fourier head learn unbounded values. We have added an explanation about this in Section 2.4 of the updated manuscript.**
>
> However, even if we decide to use a bounded domain (as is the case in all the experiments in the manuscript), this still isn’t an issue, as evidenced by the original Chronos model’s SOTA time series forecasting accuracy, with datasets which are not a priori bounded within some finite range. Those authors modeled unbounded sequences using a combination of two techniques: 1) normalizing the time series so the individual values are small, and 2) ensuring that the next-token distribution is defined over a sufficiently wide interval.
>
> In more detail: in the Chronos paper, the authors normalize the time series so that the mean of the absolute value in the historical context is equal to 1; this ensures that the time series values are small. Then, the architecture defines the range of tokens as equally spaced inside of $[-15, 15]$; this ensures that the model is capable of learning from the examples with significant trend components. With this normalization and tokenization strategy, the Chronos model is SOTA because it turns out that even time series with aggressive trend components are normalized to values that can be forecasted accurately by the model.

---

> ### Author Response · Authors · 2024-11-21
> **Author response**
>
> ## Q5: to what extent is it a weakness of the Fourier head that the user must select hyperparameters?
>
> **We have included additional experimentally-backed details to the manuscript to make it easier for the user to select hyperparameters**. Thank you for giving us the opportunity to clarify this point. In short:
>
> * When picking *regularization strength*–we find that in the low-frequency domain (i.e. frequencies in the single digits) using $\gamma=0$ works best, and in the high-frequency domain (i.e. greater than 10 frequencies) using $\gamma=10^{-6}$ works best.
>
> *  When picking *Fourier frequencies*–our Decision Transformer results show that the model is reasonably robust to the choice of the number of frequencies. For example, our latest Decision Transformer results [(link to graph)](https://drive.google.com/file/d/1TeWuaxUyFN76oaqGDzYZ9wn4Ggq3Ohj1/view?usp=drive_link) show that the Fourier agent obtains higher returns than the Linear agent, irrespective of the quantity of Fourier frequencies. Meanwhile, for more complex problems such as zero-shot probabilistic time-series forecasting, one may prefer to use more frequencies as this results in a Fourier head with more modeling power. This is stated in our Theorem 3.3 (Fourier head scaling law) and demonstrated in Table 3.

---

> ### Author Response · Authors · 2024-11-21
> **Author response**
>
> ## Q6: how were frequencies chosen for the time series experiments?
>
> We didn’t use any clever procedure for selecting the Fourier frequencies for the time series experiments. We just started by sweeping over powers of $2$ to be efficient, because the Fourier head scaling law indicates that you might want to try up to $\mathrm{n_bins} / 2 = 2048.$

---

> ### Author Response · Authors · 2024-11-21
> **Author response**
>
> ## Q7: exploring bases other than Fourier basis
>
> We want to acknowledge that there may be other bases that perform equally well to the Fourier basis. Our paper is a first step towards exploring this direction, and we expect that our results would inspire follow-up works that explore different bases. However, at a practical level, the Fourier basis is more computationally tractable (i.e. fewer FLOPS) and more stable to compute than many alternatives, such as the Chebyshev basis. Thanks for asking this question.

---

> ### Author Response · Authors · 2024-11-25
> **Author follow-up**
>
> We wanted to follow up to make sure that your concerns are being properly addressed. Please let us know if there are additional questions that we can provide further information / experiments on. Thank you again for your time and effort reviewing our paper!

---

> > ### Comment · Reviewer_aPiu · 2024-11-25
> > **Rebuttal acknowledgement**
> >
> > Thank you for the detailed rebuttal. These new experiments should make the next version of the paper more convincing. I'll raise my score to support acceptance.

---

> ### Author Response · Authors · 2024-12-03
> **Following up**
>
> We just wanted to follow up, and say thank you for your effort reviewing our paper, and for your engagement throughout the rebuttal period!

---

### Official Review · Reviewer_7iR1 · 2024-11-03

**Soundness:** 2
**Presentation:** 3
**Contribution:** 2
**Rating:** 5
**Confidence:** 4

**Summary:**

The paper introduces a novel Fourier head for large language models (LLMs), designed to improve the modeling of continuous structures in non-linguistic tokens, such as decision-making sequences in games and time series forecasting

**Strengths:**

Fourier head allows LLMs to better capture continuous structures in non-linguistic tokens, addressing the limitation in traditional models that use softmax over discrete bins.  The authors provide both theoretical justifications and empirical analysis.

**Weaknesses:**

1. The author posits that Fourier Head can endow the model with a continuity prior, , which can be described as semantic adjacency. However, since LLMs inherently incorporate attention mechanisms that aggregate tokens with higher similarities, the contribution of the Fourier Head seems incremental.

2. Regarding the time series prediction section, the author has employed the T5 architecture, yet the baseline comprises only this architecture, which is overly simplistic. There is a significant body of work on time series LLMs currently, with most eventually employing a linear layer (could also be replaced with a Fourier head), such as TimeLLM GPT4TS[1,2]. I believe the author needs to further supplement the experiments.

3. Additionally, I think the effectiveness of the Fourier Head may stem from its ability to analyze input frequency and amplitude through Fourier series. The author should consider comparing methods that are based on decoupling[3].

[1]Jin M, Wang S, Ma L, et al. Time-llm: Time series forecasting by reprogramming large language models[J]. arXiv preprint arXiv:2310.01728, 2023.

[2]Zhou T, Niu P, Sun L, et al. One fits all: Power general time series analysis by pretrained lm[J]. Advances in neural information processing systems, 2023, 36: 43322-43355.

[3]Cao D, Jia F, Arik S O, et al. TEMPO: Prompt-based Generative Pre-trained Transformer for Time Series Forecasting[C]//The Twelfth International Conference on Learning Representations.

**Questions:**

I am unclear about the organization of the paper, such as why the related work is placed in the latter half.

---

> ### Author Response · Authors · 2024-11-21
> **Author response**
>
> Thank you for the thoughtful feedback! Below we provide detailed answers to your questions. We have also uploaded a revised manuscript, with changes written in blue.

---

> ### Author Response · Authors · 2024-11-21
> **Author response**
>
> ## Q1: the Fourier head is incremental because LLMs inherently aggregate tokens with higher similarities
>
> The Fourier head’s job is to ensure that neighboring tokens have similar likelihoods during the next-token prediction step, and this is not guaranteed by LLMs. In fact, in the paper we provide examples of cases where LLMs fail to have neighboring tokens having similar likelihoods.
>
> More precisely: if you enumerate the $m$ tokens $t_1, t_2, …, t_m$, then the Fourier head ensures that the learned next-token softmax distribution satisfies the property that the likelihood of $t_i$ is close to the likelihood of $t_{i+1}$. We demonstrate in our experiments that a general attention-based transformer with a linear classification head doesn’t satisfy this same property. We illustrate this in a figure in the paper:
>  [(link to graph)](https://drive.google.com/file/d/1GNbjx3DyQhea_pRCmhLkrs8NLs6ldeLq/view?usp=drive_link).
>
> This experiment demonstrates that an LLM with a standard classification head outputs a “jagged” next-token distribution, whereas the Fourier classification head outputs a “smooth” next-token distribution.

---

> ### Author Response · Authors · 2024-11-21
> **Author response**
>
> ## Q2: the paper should include experimental results for the TimeLLM and GPT4TS models
>
> Thank you for highlighting the TimeLLM and GPT4TS papers. In the updated manuscript we have added citations to them in the related works section. **However, we don’t believe that a direct comparison to those models is relevant, because those models require in-domain training, whereas Chronos does zero-shot domain transfer.**
>
> More precisely, both the models TimeLLM and GPT4TS require fine-tuning and testing on each dataset separately. In contrast, Chronos is pretrained on a single dataset consisting of many real and synthetic time series, and it is designed to be evaluated on domains unseen during training. In our paper, we evaluate Chronos on 20 benchmark datasets unseen during training, without any additional dataset specialization or fine-tuning.
>
> Additionally, we would like to emphasize that the goal of the paper is not to present the Fourier head as a tool just for time series, but rather as a tool for any domain where a classification head can benefit from having more continuous structure. Towards this, we would like to highlight the diverse types of transformer-based models that we used to study the Fourier head–our time series experiments use an encoder-decoder T5 architecture, our reinforcement learning experiments use a decoder-only GPT-class architecture, and our audio toy example uses an audio-spectrogram transformer.

---

> ### Author Response · Authors · 2024-11-21
> **Author response**
>
> ## Q3:  the paper should include a comparison with models using decoupling such as TEMPO
>
> Thank you for bringing the TEMPO paper to our attention. Since their paper also studies how to perform time series forecasting using an LLM, we have added a citation to TEMPO in our related works section. However, we believe TEMPO is not directly comparable to the Fourier head, for the following reasons:
>
> * TEMPO employs an STL decomposition as a preprocessing step for improved training on time series data using an LLM, arguing that a transformer’s self-attention mechanism is not guaranteed to be able to disentangle the trend and seasonality components (their Theorem 3.1).
> * TEMPO proposes novel prompting techniques to improve performance.
>
> Both of these contributions are not comparable to our Fourier head since we propose an alternative way to improve the classification layer in the transformer, which is not related to either preprocessing or prompting.

---

> ### Author Response · Authors · 2024-11-21
> **Author response**
>
> ## Q4: why is the paper organized this way (specifically, with related works near the end)?
>
> We ultimately decided that we wanted the paper to flow from:
>
> 1. Motivating the desire to put a continuous structure over the next-token distribution (section 1, introduction)
> 2. Introducing our proposed method for putting a continuous structure over the next-token distribution (section 2, Fourier head)
> 3. Providing theoretical evidence that the Fourier head is capable of modeling complicated densities, explain the tradeoffs involved (section 3, Theory)
> 4. Demonstrating the Fourier head’s modeling capability in a low-dimensional intuitive example (section 4, toy example)
> 5. Demonstrating the Fourier head’s modeling capability in a large scale example (section 5, offline RL)
> 6. Demonstrating the Fourier head’s modeling capability in another large scale example (section 6, zero-shot probabilistic time series forecasting)
>
> We reasoned that putting the related works between the motivation section, and the introduction of the Fourier head method, would break up this flow, so we opted to put it in the last section before the conclusion. Not all of us liked this, so our compromise was to have a “minimal” related works explanation embedded in the introduction; this is when we discussed the Decision Transformer, Chronos, and other instances where it could be beneficial to learn next-token distributions with a continuous structure. We’re very open to alternative ways of framing the story though, did you have any suggestions?

---

> ### Comment · Reviewer_7iR1 · 2024-11-25
>
> In the abstract, the author claims that "we can easily substitute for any linear layer if we want the outputs to have a more continuous structure." However, the output linear layer of methods like TimeLLM and GPT4TS could also be replaced with a Fourier head. I believe that it is not feasible to simply substitute Fourier head in TimeLLM and GPT4TS because they do not reconstruct the continuous vocabulary. (It is worth noting that LLM-based methods like TimeLLM and GPT4TS are applicable to zero-shot forecasting tasks, as demonstrated in Section 4.8 of GPT4TS. Although the datasets used differ, these LLM-based methods can perform well in out-of-domain tasks.)
>
> Besides LLM-based methods, other TS foundation models such as UNITS[1] and MOMENT[2] cannot replace their output layers with Fourier head either. The author does not adequately explain how to implement Fourier head in these scenarios, which I think limits their usability.
>
> Chronos is also a TS foundation model, and compared to LLM-based methods, TS foundation models require significant computational resources as they need to trainning from scratch or fully tuning on large datasets. Their broad applicability remains unproven (whereas LLM-based methods could be quickly trained and validated). If constructing an ordered vocabulary is an essential part of the process, then this disadvantage cannot be ignored.
>
> [1] Gao S, Koker T, Queen O, et al. Units: Building a unified time series mode. NeurIPS2024
>
> [2] Goswami M, Szafer K, Choudhry A, et al. MOMENT: A Family of Open Time-series Foundation Models[C]//Forty-first International Conference on Machine Learning.

---

> > ### Author Response · Authors · 2024-12-02
> > **Quick follow up**
> >
> > Dear reviewer 7iR1,
> >
> > Thank you again for your constructive feedback and your engagement with our response!
> >
> > As the discussion phase is ending soon, we would like to quickly follow up with you and see if our responses on _when to use Fourier head_, _time-series forecasting baselines and Chronos_, and _the broader impact of Fourier head_ help address your concerns. We look forward to your comment and will try our best to address any remaining questions you might have!
> >
> > Best,
> >
> > The authors

---

> ### Author Response · Authors · 2024-11-25
> **Author response**
>
> We appreciate the reviewer’s engagement with us and would like to put the contributions of our Fourier Head in proper perspective with respect to the time-series forecasting methods the reviewer brought up.
>
> 1. *When to use Fourier Head:* As illustrated in our new toy experiments (revised manuscript, Figure 3 and Table 6), it is desirable to use the Fourier Head when we aim to model **complex** probabilistic distributions without knowing a priori the family of distributions. Particularly, Figure 3 shows that when the underlying distribution follows a Beta distribution, both vanilla linear head (no inductive bias) or GMM head (wrong inductive bias) are significantly outperformed by Fourier Head. Table 6 and Figure 9 [(link to figure)](https://drive.google.com/file/d/1ikH_DDJcVCTGslCzKxvSlYL326g8tpK2/view?usp=sharing) further show that when the target distribution has multiple modes (e.g. GMM or Beta), pointwise methods regress to the mean, and fail to capture any of the modes.
>
> 2. *TS forecasting baselines:* To the best of our knowledge, GPT4TS, UNITS, and MOMENT are all pointwise forecasting models; they are hence “incompatible” with Fourier Head, which is designed to model (discretized) distributions. However, we believe the quoted statement from us remains true: one just needs to change both the “head” and the training objective (away from pointwise estimation), which we believe may lead to improved empirical performance as discussed in our response above. As the reviewer could see, these improvements are orthogonal to the design choices made by GPT4TS, UNITS, or MOMENT, meaning our contributions are complementary, not competing against these prior works.
>
> 3. *Validity of the Chronos implementation:* Chronos is a state-of-the-art probabilistic TS forecasting model recently accepted by the TMLR journal. We integrated the Fourier Head implementation into Chronos for its competitive performance, as well as open-source data preparation and training pipelines. We acknowledge that compared with TimeLLM or other similar methods, Chronos requires an additional training / fine-tuning stage. We would also like to clarify that the vocabulary construction process is independent from the requirement of “training from scratch”: Chronos actually explored fine-tuning with pre-trained LLM weights but observed that it has marginal impact on the performance. Additionally, updating the “vocabulary”, or even incorporating continuous features, is a standard practice, as exemplified by numerous multimodal LLM works, such as LLAVA. We hence believe this should not be considered as a fundamental disadvantage.
>
> 4. *Broader impact of Fourier Head:* Finally, we would like to clarify that we have demonstrated broader applicability of the Fourier Head beyond the time-series forecasting applications. We invite the reviewer to check our decision transformer experiments, especially the newly added evaluations [(link to figure)](https://drive.google.com/file/d/1TeWuaxUyFN76oaqGDzYZ9wn4Ggq3Ohj1/view?usp=drive_link) where Fourier Head demonstrates consistent and significant performance improvements.
>
> Lastly–please let us know if there are additional questions that we can provide further information / experiments on. Thank you again for your time and effort reviewing our paper!

---

> ### Author Response · Authors · 2024-12-03
> **Following up**
>
> We wanted to follow up, and check if our answers have satisfied your concerns. Thanks again for your effort reviewing our paper, and for your engagement throughout the rebuttal period!

---

### Official Review · Reviewer_qzaH · 2024-11-04

**Soundness:** 2
**Presentation:** 2
**Contribution:** 2
**Rating:** 6
**Confidence:** 3

**Summary:**

The paper proposes a Fourier Head based on the Fourier series as a replacement for the usual linear classification head to induce continuous densities across the class IDs. It presents a theoretical analysis of the expressiveness and smoothness trade-off as the number of frequencies increases and empirically shows the advantage of the Fourier Head over the conventional linear head in tasks with output classes with a continuous structure.

**Strengths:**

- The proposed Fourier Head layer is well-motivated in domains where classes have a continuous structure
- The method is straightforward and clearly explained
- Visualizations clearly demonstrate the advantage of the Fourier head on toy problems in learing continuous densities
- Experiments in RL and time-series show the Fourier head can improve performance in non-toy settings

**Weaknesses:**

- As the paper is focused on improving LLM's ability to model numerical values, there are important related works that explore alternative ways of extracting continuous probabilistic predictions from LLMs over numerical data [1, 2], which are worth discussing. These methods use a hierarchical representation of the numerical values, encouraging nearby values to have similar densities, as they do not correspond to independently predicted classes. These methods therefore do not have the limitations of "not consider any continuous structure that resides among the tokens", which the Fourier head claims to address.
- Similar to methods based on classification over binned values, the Fourier head can only represent a finite range of values. Methods like [1, 2] in principle do not have this issue.
- The advantage of using the Fourier head seems most significant with small models trained on limited data. At a large scale, the model should be able to learn the continuous structure in the output classes, diminishing the benefit of using the Fourier head. It would be useful to show how the benefit of replacing the linear head with the Fourier scale with training data and model size, such as for Chronos models of different sizes.

[1] Gruver et al. 2023. Large Language Models Are Zero-Shot Time Series Forecasters

[2] Requeima et al. 2024. LLM Processes: Numerical Predictive Distributions Conditioned on Natural Language

**Questions:**

- Can you provide more details on the Decision Transformer and Chronos experiments? How did you choose the size of the models, and how long to train?
- Can you show how the benefit of using the Fourier head varies as the model size or amount of training data increases? That is, does the benefit of the Fourier head persist at scale or does it vanish?

---

> ### Author Response · Authors · 2024-11-21
> **Author response**
>
> Thank you for the thoughtful feedback! We were able to produce the additional results that you asked for, summarized here:
> * **We conduct an additional ablation which demonstrates that the benefit of the Fourier head persists as the *dataset size* scales.**
> * **We conduct an additional ablation which demonstrates that the benefit of the Fourier head persists as the *model size* scales.**
>
> Below, we provide more details, as well as question-specific responses. We have also uploaded a revised manuscript, with changes written in blue.

---

> ### Author Response · Authors · 2024-11-21
> **Author response**
>
> ## Q1: alternative ways of extracting continuous probabilistic predictions from LLMs over numerical data
>
> Thank you for mentioning the papers [1] Gruver et al and [2] Requeima et al, we have added their citations to the discussion in the related works section. However, we note that the Fourier head avoids a practical limitation from both of those methods. **In short, the Fourier head’s design ensures that the categorical distribution which is learned during training is exactly the same distribution which is sampled from during evaluation. This is not the case for the methods in [1] and [2].**
>
> In more detail–in the methods from [2], the authors state the following limitation of their work: *It must be noted that this approach does not guarantee that P(12) yields the mass assigned by the LLM to values in the bin [12, 13). However, we note that our method defines a valid predictive distribution, and we empirically observed that our predictive distribution closely matches the sampling distribution.* In the appendix, they share images that show how the distributions are visually similar, but not identical. In contrast: in the Fourier head, the mass assigned to any range of numbers is exactly equal to the integral. This is a theoretical and practical benefit of modeling tokens over numerical distributions continuously using a smooth density (as in the Fourier head) rather than learning a discretized hierarchical softmax (as in [1] and [2]).

---

> ### Author Response · Authors · 2024-11-21
> **Author response**
>
> ## Q2: the Fourier head can only represent a finite range of values
>
> In practice, we find that this is not an issue. **If needed, a tanh reparameterization can be used to map the domain $[-1,1]$ to the real line. This would let the Fourier head learn unbounded values. We have added an explanation about this in Section 2.4 of the updated manuscript.**
>
> However, even if we decide to use a bounded domain (as is the case in all the experiments in the manuscript), this still isn’t an issue, as evidenced by the original Chronos model’s SOTA time series forecasting accuracy, with datasets which are not a priori bounded within some finite range. Those authors modeled unbounded sequences using a combination of two techniques: 1) normalizing the time series so the individual values are small, and 2) ensuring that the next-token distribution is defined over a sufficiently wide interval.
>
> In more detail: in the Chronos paper, the authors normalize the time series so that the mean of the absolute value in the historical context is equal to 1; this ensures that the time series values are small. Then, the architecture defines the range of tokens as equally spaced inside of $[-15, 15]$; this ensures that the model is capable of learning from the examples with significant trend components. With this normalization and tokenization strategy, the Chronos model achieved SOTA accuracy because it turns out that even time series with aggressive trend components are normalized to values that can be forecasted accurately by the model.

---

> ### Author Response · Authors · 2024-11-21
> **Author response**
>
> ## Q3: how does the Fourier head’s performance scale with training data and model size?
>
> Thank you for suggesting adding these valuable experiments, which we have added to the revised draft. We summarize the results of our additional ablations here, and also include links to the graphs:
>
> * Ablation study #1: We show the Fourier head is better at learning high-quality next action distributions than the Decision Transformer with a Linear head *across all dataset sizes*. [(link to graph)](https://drive.google.com/file/d/16qJsSBJ9xwT6PiqYqYco6pBkUiHaXX9x/view?usp=sharing)
> * Ablation study #2: We show the Fourier head is better at learning high-quality next action distributions than the Decision Transformer with a Linear head *across all model sizes*. [(link to graph)](https://drive.google.com/file/d/1-mnyWD4F1Rqgj3-xSVenRp17UWGAMGU1/view?usp=drive_link)
>
> **These results demonstrate that the benefits of the Fourier head indeed persist with scale.**

---

> ### Author Response · Authors · 2024-11-21
> **Author response**
>
> ## Q4: providing more details on the Decision Transformer and Chronos experiments
>
> For both Decision Transformer and Chronos, we followed the same training recipes from their original implementations. At your suggestion, we’ve added brief summaries of them to the paper. We will share some details here as well.
>
> * *Our Decision Transformer experiments:* following the original implementation, we trained on 500k transitions observed by a DQN agent during training, for 5 epochs. We trained on the same model size as the original implementation (a GPT-1 model with approx. 2.012M parameters) which takes about four hours on a single GPU.
>
> * *Our Chronos experiments:* following the original implementation, we trained for 200k steps, on the same model size as the original implementation (a T5 model with approx. 20M parameters) which takes just under 48 hours on 8 GPUs.

---

> ### Author Response · Authors · 2024-11-25
> **Author follow-up**
>
> We wanted to follow up to make sure that your concerns are being properly addressed. Please let us know if there are additional questions that we can provide further information / experiments on. Thank you again for your time and effort reviewing our paper!

---

> > ### Comment · Reviewer_qzaH · 2024-11-26
> >
> > I thank the authors for the clarifications and additional scaling results. Regarding the range of representable values, I agree in principle this can be addressed by compactifying the domain. But the more relevant question is whether the inductive bias of the model is natural for representing an unbounded range of values once a tanh like transformation is involved.
> >
> > It would be great to show similar scaling experiments for other tasks such as the Chronos experiment, where both the model and data size are much larger than the decision transformer experiment (e.g. model only has 2M parameters) and arguably more representative of the practical setting.

---

> ### Author Response · Authors · 2024-11-26
> **Author response**
>
> In the original Fourier Basis Density Model paper, the authors showed that the Fourier inductive bias is indeed natural for representing an (a priori) unbounded range of values once a tanh transformation is involved. In more detail–one of their modeling tasks involves learning a mixture of 25 Gaussians using a Fourier density with the tanh reparameterization, where most of the probability density is concentrated inside the range $[-10, 10]$. They find that the reparameterized Fourier density model can accurately learn the density, while capturing more modes and using fewer parameters than alternative models. (In case you’re curious,  [here is a link](https://drive.google.com/file/d/1nDU994lZkMQfDNn9yO0tArk6XTQXWcdh/view?usp=sharing) to an illustration of this example from their paper.)
>
> We agree with the reviewer that results on scaling to even larger data size and model size would be interesting. We would like to clarify that our Chronos experiments, which demonstrate the effectiveness of Fourier Head, are already “large-scale”. The model has 20M parameters, and is pre-trained on over 10 million data points (the same setup as used by the Chronos paper). Further scaling up the Chronos model size and data size by orders of magnitude would be beyond the scope of our academic training size. Additionally, the scaling behavior of time-series forecasting models is on its own an active research area, we refer the reviewer to recent papers such as [(Shi et al., NeurIPS 2024)](https://arxiv.org/abs/2405.15124) on this topic.

---

> > ### Comment · Reviewer_qzaH · 2024-11-26
> >
> > I thank the authors for linking to the experiment from the Fourier basis density model paper. I'm less concerned with representing unbounded values with the Fourier head now.
> >
> > I believe it is not unreasonable to train larger models on more data than what is done in the Chronos experiment. For example, the 124M GPT-2 small model is frequently used to validate new methods within an academic compute budget. In addition to training at larger scales, one can also train at smaller scales to extrapolate a trend.

---

> ### Author Response · Authors · 2024-11-28
> **Author response**
>
> We have conducted an additional study to analyze whether dataset size has any effect on the relative performance of the Linear head and the Fourier head for Chronos. Our results show that, across dataset sizes, the Fourier head indeed yields more accurate forecasts than the Linear head. We have updated the manuscript with these changes. (And [here is a link](https://drive.google.com/file/d/1nMR6dNN_s3eJZ1QXFc1wAXjL9odXvM4v/view?usp=sharing) to the figure that we added to the paper.) Thanks for the suggestion!
>
> Additionally, we have started running experiments on scaling model size for Chronos, and we are hoping to finish them before the end of the extended rebuttal period.

---

> ### Author Response · Authors · 2024-12-02
> **Author response**
>
> As you requested, we have conducted an additional ablation study to analyze whether *model size* has any effect on the relative performance of the Linear head and the Fourier head for the Chronos time series forecasting task. Our results show that, across model sizes, the Fourier head indeed yields more accurate forecasts than the Linear head. It looks like we are not allowed to update the manuscript in OpenReview anymore, so [here is a link](https://drive.google.com/file/d/1ZsGoE1NN1GufSni9TP1MsSSJv2jQ3Oqs/view?usp=sharing) to the paper updated with these changes. (And [here is a link](https://drive.google.com/file/d/1S7bsytI1oN8W27medI9Po1bcZBjCMpbs/view?usp=sharing) to just the figure that we added to the paper.)
>
> Thanks for your patience during this rebuttal period while we ran our experiments, and thanks for suggesting that we run this experiment. Please let us know if you have any other questions!

---

> ### Author Response · Authors · 2024-12-03
> **Following up**
>
> We wanted to follow up, and check if the additional ablation studies on scaling model size and dataset size for Chronos satisfied your concerns. Thanks again for your effort reviewing our paper, and for your engagement throughout the rebuttal period!

---

### Author Response · Authors · 2024-11-21
**Author Response Summary**

We thank all the reviewers for the detailed feedback. We have conducted the additional requested experiments and incorporated the results in the paper:

* In response to Reviewer qzaH’s question about whether the benefit of the Fourier head persists at various *data scales*, we conducted an additional ablation study which demonstrates that the Fourier head consistently outperforms the Linear head baseline, across dataset sizes.
* In response to Reviewer qzaH’s question about whether the benefit of the Fourier head persists at various *model scales*, we conducted an additional ablation study which demonstrates that the Fourier head consistently outperforms the Linear head baseline, across model sizes.
* In response to Reviewer aPiu’s suggestion that we demonstrate further empirical impact with additional RL experiments, we provide additional RL experiments on three more Atari games, where the Fourier head agent consistently obtains higher returns than the linear head agent.
* In response to Reviewer aPiu’s suggestion that we compare the Fourier head to more baselines, we provide an additional baseline by constructing a Gaussian Mixture Model-based classification head, as well as a more challenging synthetic dataset using the Beta distribution. Our results on this extended baseline demonstrate that the Fourier head’s flexibility is needed to model distributions which are even slightly more complicated than Gaussians.
* In response to Reviewer WB6H's suggestion that we consider a regression baseline, we expanded the toy example experiments with a Pointwise Regression head. We demonstrate that this new model “regresses to the mean” because of the probabilistic nature of the dataset, and as a result it doesn’t have any advantage over the classification heads, since the classification heads model the probabilistic datasets by directly modeling the underlying probability distributions.

In terms of writing updates to the manuscript (written in blue in the updated manuscript):

* In response to Reviewers qzaH and 7iR1, we added references to more LLM time series papers.
* In response to Reviewer qzaH, we added more details on the Decision Transformer and Chronos experiments to the appendix.
* In response to Reviewer WB6H, we clarified why quadratic fourier regularization is useful even when the Fourier series is truncated, and we fixed some minor typos.
* In response to Reviewer aPiu, we added more details about how to choose hyperparameters.
* In response to Reviewers aPiu and qzaH, we added details about how the Fourier head can be adapted to model unbounded values.

---

### Comment · Area_Chair_oxft · 2024-11-24

Dear Reviewers,

This is a gentle reminder that the authors have submitted their rebuttal, and the discussion period will conclude on November 26th AoE. To ensure a constructive and meaningful discussion, we kindly ask that you review the rebuttal as soon as possible and verify if your questions and comments have been adequately addressed.

We greatly appreciate your time, effort, and thoughtful contributions to this process.

Best regards,
AC

---

### Meta-Review · Area_Chair_oxft · 2024-12-28

**Metareview:**

This paper proposes a new mechanism to model discrete distributions using Fourier representations. The motivation is to better model the underlying low-frequency signals in the Fourier space. Empirical validation is done in two domains for offline RL and time series foundation models. The reviewers mainly appreciated the novelty of the approach and the ablations for the design choices. Their main concerns were regarding the limited empirical validation both in terms of baselines and additional environments for testing.

**Additional Comments On Reviewer Discussion:**

The paper was borderline with concerns ranging from clarity of exposition to limited experimentation. The authors were able to address some of these limitations. Some choices are still a little ad-hoc: for example, why focus on offline RL and time-series modeling, and not language modeling itself? The latter is the prime example of discrete distributions. Similarly, the Atari experiments are focussing on a non-overlapping set of environments from the original paper. Overall, I think more could have been done by the authors in either demonstrating universal utility (which I do not think is the case based on current empirical evidence), or focussing deeper in a specific domain (either offline RL, time-series modeling) and doing more extensive experimentation to identify the exact reasons for improved performance in those domains, or add additional benchmarks.

---

### Decision · Program_Chairs · 2025-01-22

Accept (Poster)